# Deciphering the role of cis-regulatory elements and TFAP2C in the activation of zygotic *Sox2* expression in mouse preimplantation embryos

Jaehwan Kim[1,*], Chad S. Driscoll[1], Lijia Li[2], Catherine A. Wilson[1], Wei Xie[2] and Jason G. Knott[1,‡]

## ABSTRACT

Cell fate decisions in preimplantation embryos require the coordinated expression of pluripotency and lineage-specific transcription factors. SOX2 represents the first pluripotency regulator for which expression is restricted to the inside cells of mouse preimplantation embryos. However, the genetic mechanisms that activate the expression of zygotic *Sox2* are poorly understood. Here, we report that *Sox2* expression in mouse embryos is controlled by the actions of key cis-regulatory elements, including a proximal promoter and super enhancer. We show that TFAP2C, a key trophoblast lineage regulator, binds to the *Sox2* proximal promoter to activate its expression. Lastly, we provide evidence that TFAP2C and the HIPPO signaling pathway cooperatively regulate *Sox2* expression. In summary, this work has important implications for understanding how conventional trophoblast transcription factors, such as TFAP2C, contribute to the activation of early pluripotency genes to facilitate divergent cellular states that support lineage formation.

KEY WORDS: Preimplantation embryo, *Sox2* regulatory regions, TFAP2C, Pluripotency, HIPPO signaling

## INTRODUCTION

In mammals, life begins as a zygote that has the remarkable ability to differentiate into all of the specialized cells in the embryo proper and extra-embryonic tissues. The ability of embryonic cells to differentiate into the future embryonic versus extra-embryonic tissues is acquired during the first and second cell fate decisions. The first cell fate decision encompasses the formation of the inner cell mass (ICM) and multipotent trophectoderm (TE) (i.e. future placenta) during the morula-to-blastocyst transition (Cockburn and Rossant, 2010). The second cell fate decision involves the segregation of the blastocyst ICM into the pluripotent epiblast and multipotent primitive endoderm (PE) lineages. The epiblast gives rise to all of the somatic cells and germ cells in the embryo, while the

PE lineage contributes to the formation of the yolk sac (Cockburn and Rossant, 2010).

Many different developmental transcription factors (TFs) have been shown to be important for these initial cell fate decisions and the acquisition of pluripotency and multipotency cellular states (Karasek et al., 2020; Chowdhary and Hadjantonakis, 2022). Of particular interest is SOX2, an SRY-box TF that represents the first pluripotency factor in mice that is specifically expressed on the inside cells of morulae, which contribute to the future blastocyst ICM (Guo et al., 2010; Wicklow et al., 2014). The onset of zygotic *Sox2* expression and its localization are negatively controlled by TEAD4 and YAP1, the downstream effector of the evolutionarily conserved HIPPO signaling pathway (Wicklow et al., 2014; Frum et al., 2018, 2019). Genome-wide binding analysis of SOX2 in blastocysts and early postimplantation embryos revealed that SOX2 exerts multifaceted chromatin interactions at target gene enhancers to regulate various pluripotency states (i.e. pre-, naïve and formative pluripotency) (Li et al., 2023). Importantly, loss-of-function studies have demonstrated that zygotic SOX2 is required for the proper segregation of the blastocyst ICM into the epiblast and PE lineages, and for maintenance of pluripotency in the epiblast (Wicklow et al., 2014; Mistri et al., 2018).

Despite our understanding of the role of SOX2 in mouse pre- and postimplantation development and its negative regulation by the HIPPO pathway, there are significant gaps in our knowledge of how *Sox2* expression is activated during early embryogenesis. Notably, the cis-regulatory elements that control zygotic *Sox2* expression, as well as the developmental TFs that contribute to its activation during early embryogenesis, remain unknown. Here, we report that the *Sox2* proximal promoter and components of its distal super enhancer are required for *Sox2* transcriptional activation. Furthermore, we demonstrate that transcription factor AP2 gamma (TFAP2C), a well-established regulator of the trophoblast lineage (Choi et al., 2012; Cao et al., 2015; Kuckenberg et al., 2010) and 8-cell bipotency, (Li et al., 2024; Zhu et al., 2024), binds to the proximal promoter and activates *Sox2* expression between the 8-cell and morula stages. Lastly, we provide evidence that TFAP2C and the HIPPO signaling pathway cooperatively regulate *Sox2* expression at the morula stage.

## RESULTS

### Identification of key cis-regulatory elements required for zygotic *Sox2* expression in mouse embryos

The underlying cis-regulatory elements that are required for *Sox2* transcription in mouse preimplantation embryos are largely unknown. Studies in mouse neuronal cells and embryonic stem cells (ESCs) have revealed that *Sox2* expression is differentially regulated by specific proximal and distal enhancers. For example, in neurons, the proximal *Sox2* regulatory regions (SRRs), SRR1 and

[1]Developmental Epigenetics Laboratory, Department of Animal Science, Reproductive and Developmental Sciences Program, Michigan State University, East Lansing, MI 48824, USA. [2]Center for Stem Cell Biology and Regenerative Medicine, MOE Key Laboratory of Bioinformatics, New Cornerstone Science Laboratory, School of Life Sciences, Tsinghua University, Beijing 100084, China.
*Present Address: Primate Resource Center, Korea Research Institute of Bioscience and Biotechnology (KRIBB), Jeongeup 580883, Republic of Korea.

‡Author for correspondence (knottj@msu.edu)

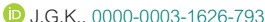 J.G.K., 0000-0003-1626-7933

SRR2, are required for *Sox2* expression (Sikorska et al., 2008; Lopez-Juarez et al., 2012), whereas in ESCs SRR1 and SRR2 are not necessary, but a downstream super enhancer consisting of SRR107 and SRR111 is essential (Zhou et al., 2014; Li et al., 2014). To address which cis-regulatory elements are required for *Sox2* expression in preimplantation embryos, we generated a fluorescence reporter assay that has been utilized in previous studies (Rayon et al., 2014). This reporter construct serves as a tool to screen for gene promoters and enhancers that are activated during early development. Using PCR, we isolated SRR1, SRR2, and SRR107 and SRR111, two components of the *Sox2* super enhancer. We also isolated a ~1.6 kb promoter/partial-coding region (CDS) of *Sox2* that spans the transcriptional start site (Fig. 1A, Fig. S1). Each of the SRRs were then individually cloned upstream of a histone H2b (H2B)-red fluorescence protein (RFP) vector containing the minimal β-actin promoter (Fig. S1). The *Sox2* promoter/partial-CDS region was inserted into the reporter construct without the minimal β-actin promoter. PCR was then used to produce linear SRR/promoter H2b-RFP constructs for microinjection. A construct containing H2b-RFP and the minimal β-actin promoter alone was used as a negative control.

The activities of the five selected SRR/promoter H2b-RFP constructs were tested individually by pronuclear injection into 1-cell embryos (Fig. 1B). Embryos were cultured to the morula stage, when SOX2 protein is first expressed, and confocal immunofluorescence (IF) analysis was performed using an anti-RFP antibody. The relative nuclear RFP intensity was measured in each group of embryos and compared to DAPI staining. Consistent with *Sox2* regulation in ESCs, the proximal enhancers SRR1 and SRR2 were not activated in preimplantation embryos (Fig. 1B,C). As expected, embryos injected with H2b-RFP alone did not exhibit RFP fluorescence (Fig. 1B,C). Conversely, embryos injected with the *Sox2* promoter/partial-CDS and SRR111 exhibited a significant increase in RFP fluorescence compared to embryos injected with the H2B-RFP construct ($P<0.0001$; Fig. 1B,C). Embryos injected with SRR107 exhibited increased levels of RFP fluorescence, although the difference was not statistically significant compared to the control ($P=0.09$; Fig. 1B,C). These results indicate that the promoter/partial-CDS and super enhancer (i.e. SRR111) are involved in the activation of *Sox2* expression during early embryogenesis in mice.

Because the reporter assay merely acts as a screening tool to identify regulatory regions required for *Sox2* expression and the expression did not completely mimic endogenous SOX2 patterning, we utilized CRISPR/Cas9 to delete the endogenous SRR107 and SRR111 enhancers in preimplantation embryos. Note that we did not use CRISPR to delete the *Sox2* promoter/partial-CDS region because ablation of this region would likely disrupt the assembly of the general transcription machinery and block the initiation of transcription. Thus, we focused on the two super enhancer regions (SRR107 and SRR111), which are known to be important for the regulation of *Sox2* in ESCs. Three pairs of targeting sgRNAs were designed to delete either SRR107 or SRR111 alone or the entire super enhancer control region (SCR). The CRISPR-deletion efficiency in single embryos was confirmed by PCR genotyping using outside and inside primers at the targeted regions (Fig. S2). This analysis revealed homozygous targeting in most embryos (11/15; 73%). To test the effects of the SRR107/111 deletion on *Sox2* expression, the injected embryos were cultured to the early morula stage. As a negative control, Cas9 and a non-targeting (NT) sgRNA were injected into embryos. Real-time quantitative PCR (qPCR) analysis revealed that in the SCR-, SRR107- and SRR111-targeted embryos, *Sox2* transcripts were reduced by 70,

69 and 62%, respectively, compared to embryos injected with Cas9 and the NT sgRNA ($P<0.0001$; Fig. 1D). Importantly, the requirement of SRR107 and SRR111, but not SRR1 and SRR2, in *Sox2* transcriptional activation, appears to be conserved in mouse preimplantation embryos and ESCs (Zhou et al., 2014; Li et al., 2014).

Next, we evaluated the effects of SCR, SRR107 or SRR111 deletions on SOX2 protein expression and early lineage allocation using CDX2 as a marker of the emerging TE (Fig. 1E,F). We hypothesized that a decrease in the SOX2-positive population of cells would alter the ratio of ICM and TE cells. To test this, we counted the number of SOX2- and CDX2-positive cells in early blastocysts using *z*-stack images that were generated using confocal IF. The ratios of the SOX2-positive nuclei and CDX2-positive nuclei to total DAPI-positive cells were calculated. The number of inner SOX2-positive cells decreased after ablating SCR, SRR107 and SRR111. Compared to the controls, the Cas9/sgRNA-injected embryos exhibited a reduction in the SOX2-positive to total cell ratio and an increase in the CDX2-positive cell ratio, indicative of altered lineage allocation. The SOX2-positive cell ratio for SCR-, SRR107- and SRR111-deleted embryos was 0.15, 0.24 and 0.24, respectively, compared to 0.32 in embryos injected with Cas9 and the NT sgRNA (Fig. 1E,F). The CDX2-positive cell ratio exhibited the opposite trend, showing an increase in CDX2-positive cells, with the SCR deletion having the strongest effect. In the SCR-, SRR107- and SRR111-deleted embryos, the ratio was 0.85, 0.76 and 0.76, respectively, compared to 0.68 in the controls (Fig. 1E,F). Collectively, these results demonstrate that the *Sox2* promoter/partial-CDS region and the super enhancer consisting of SRR107 and SRR111 play a crucial role in *Sox2* expression, and that disruption of SRR107/111 alters the allocation of ICM and TE cells in blastocysts. Further studies are necessary to understand the potential role of these cis-regulatory regions in early lineage formation.

### TFAP2C preferentially binds to the *Sox2* proximal promoter during preimplantation embryo development

Research from our laboratory and other groups has revealed that the trophoblast regulator TFAP2C can positively regulate both TE and pluripotency genes during early embryogenesis in mice (Choi et al., 2012, 2013; Driscoll et al., 2024a; Li et al., 2024; Zhu et al., 2024; Cao et al., 2015). These include the TE genes *Cdx2* and *Gata3*, and the pluripotency genes *Pou5f1* (also known as *Oct4*), *Nanog* and *Nr5a2*. A previous motif analysis of the *Sox2* super enhancer revealed the presence of numerous TF binding motifs (Zhou et al., 2014). We further examined SRR107/111, SRR1/2 and the promoter/partial-CDS region. We identified several putative TFAP2C binding sites within these regions through manual analysis. Accordingly, we found four motifs within SRR107/111, nine motifs within the 1.6 kb promoter/partial-CDS, and one TFAP2C motif in the SRR2 enhancer. No TFAP2C motifs were identified in SRR1 (Table S1).

To test whether TFAP2C binds to these regions, we re-analyzed a recently published CUT&RUN dataset from mouse preimplantation embryos (Li et al., 2024). This study evaluated the genome-wide binding of TFAP2C in 2-cell, 4-cell, 8-cell, blastocyst and day 6.5 postimplantation stage embryos. This analysis revealed that TFAP2C binding was highly enriched at the *Sox2* proximal promoter at the 4-cell and 8-cell stages and then was lost at the blastocyst stage (Fig. 2). Interestingly, TFAP2C binding was weakly enriched at SRR107 and SRR111 in preimplantation stage embryos, but it was greatly enriched after the blastocyst stage in day 6.5 post-implantation stage embryos when *Sox2* is expressed and functions in the trophoblast lineage (Adachi et al., 2013; Li et al., 2023)

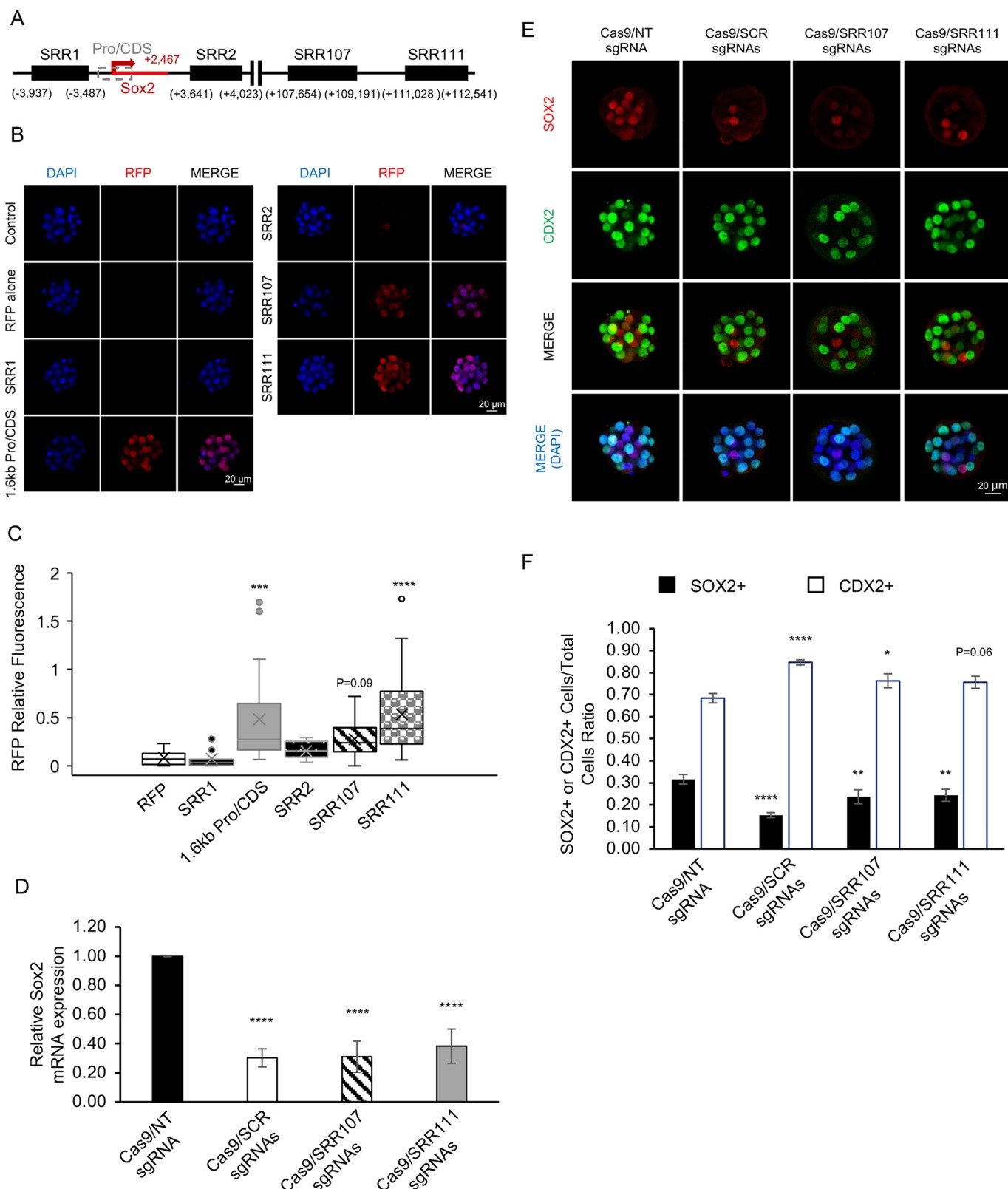

**Fig. 1.** See next page for legend.

(Fig. 2). TFAP2C binding was not observed at SRR1 and SRR2 in pre- or postimplantation stage embryos. Because the study by Li et al. (2024) did not include embryos at the morula stage, we examined a second CUT&RUN data set (Gao et al., 2024) using the

UCSC genome browser (Perez et al., 2025). This analysis revealed that TFAP2C binding was enriched at the *Sox2* proximal promoter in both 8-cell and morula-stage embryos (Fig. S3), before being lost at the blastocyst stage (Fig. 2) (Li et al., 2024). As a negative control,

**Fig. 1. The *Sox2* promoter and super enhancer are required for *Sox2* expression in mouse preimplantation embryos.** (A) Genomic locations of the *Sox2* regulatory regions (SRRs) and ~1.6 kb promoter/partial-coding region (Pro-CDS) that were tested in embryos using reporter constructs. (B) Confocal IF analysis of embryos injected with the different *Sox2* SRR and Pro-CDS reporter constructs. The analysis was done at the morula stage (E3.25). Uninjected embryos and embryos microinjected with the H2b-RFP construct alone were used as negative controls. The reporter construct activity was analyzed through H2B-RFP expression. (C) Box plot analysis of the relative nuclear H2B-RFP intensity of embryos injected with the SRR, Pro-CDS and control constructs. DAPI was used to normalize the relative RFP intensity. The bottom horizontal line of the box represents the first quartile (Q1); 25% of the data fall below this value. The top horizontal line represents the third quartile (Q3); 75% of the data fall below this value. The X inside the box represents the mean value, and the horizontal line inside the box represents the median (Q2). The interquartile range (Q3−Q1) represents the middle 50% of the data. The whiskers highlight the general range of the dataset, excluding the outliers. The dots located outside of the box indicate exceptional values (i.e. outliers) that do not fit the pattern. Three biological replicates were used for each reporter construct, with a total of 15-30 embryos per group. Values are presented as mean±s.d. \*\*\**P*<0.001, \*\*\*\**P*<0.0001 (one-way ANOVA). (D) Real-time qPCR analysis of *Sox2* transcripts in control, SCR, SRR107 and SRR111 at the morula stage (E3.25). Three biological replicates were used, with 20 pooled embryos per replicate. *Ubtf* was utilized as a normalization control. Values are presented as mean±s.d. \*\*\*\**P*<0.0001 (one-way ANOVA). (E) Confocal IF analysis of SOX2 and CDX2 in early blastocysts (E4.25). Nuclei were counterstained with DAPI (blue) to calculate the total number of cells. (F) Quantification of the SOX2 and CDX2 positive cell to total cell ratios in control, SCR-, SRR107- and SRR111-deleted embryos. Ten embryos were analyzed in each group. Values are presented as mean±s.d. \**P*<0.05, \*\**P*<0.01, \*\*\*\**P*<0.0001 (one-way ANOVA). Data analyzed with one-way ANOVA were subjected to the Dunnett's post-hoc test. Scale bars: 20 μm.

we analyzed TFAP2C binding in day 6.5 epiblasts when TFAP2C expression is downregulated (Li et al., 2024) (Fig. 2). No TFAP2C enrichment peaks were observed in these samples, confirming the specificity of the TFAP2C antibody as shown previously (Li et al.,

2024). Additionally, we evaluated histone H3 lysine 27 acetylation (H3K27ac), an epigenetic marker of active enhancers (Creyghton et al., 2010). We observed enrichment of H3K27ac at SRR107 and SRR111 in 4-cell and 8-cell stage embryos and in day 6.5 extra-embryonic ectoderm (ExE). Collectively, these results show that TFAP2C predominately binds to the *Sox2* proximal promoter between the 4-cell and morula stages of preimplantation embryo development and may represent an activator of zygotic *Sox2* expression.

## TFAP2C is a major regulator of zygotic *Sox2* expression at the 8-cell and morula stages

To understand the potential role of TFAP2C in *Sox2* expression, we first evaluated the developmental expression and localization of TFAP2C and SOX2 proteins from the 2-cell to the blastocyst stages using confocal IF analysis (Fig. S4). We also evaluated *Sox2* transcripts in oocytes and preimplantation embryos using real-time qPCR analysis (Fig. S4). TFAP2C protein was expressed and localized in the nuclei of 2-cell, 8-cell and morula-stage embryos. At the blastocyst stage, TFAP2C was downregulated in the ICM and enriched in the TE epithelium, as shown earlier (Kuckenberg et al., 2010; Choi et al., 2012). SOX2 protein was not detected at the 2-cell and 8-cell stages, but was greatly induced at the morula stage, where it was localized to a subset of nuclei in the inside cells, as previously shown by others (Wicklow et al., 2014; Frum et al., 2019). Following the morula-to-blastocyst transition, SOX2 was restricted exclusively to the ICM and was absent in the TE. Consistent with SOX2 protein expression, qPCR analysis revealed that *Sox2* transcripts were low in oocytes and 2-cell-stage embryos but were significantly induced between the 8-cell and morula stages (Fig. S4). Collectively, these analyses demonstrated that TFAP2C protein expression precedes the onset of zygotic *Sox2* transcription, indicating that TFAP2C may represent an activator of *Sox2* expression between the 8-cell and morula stages.

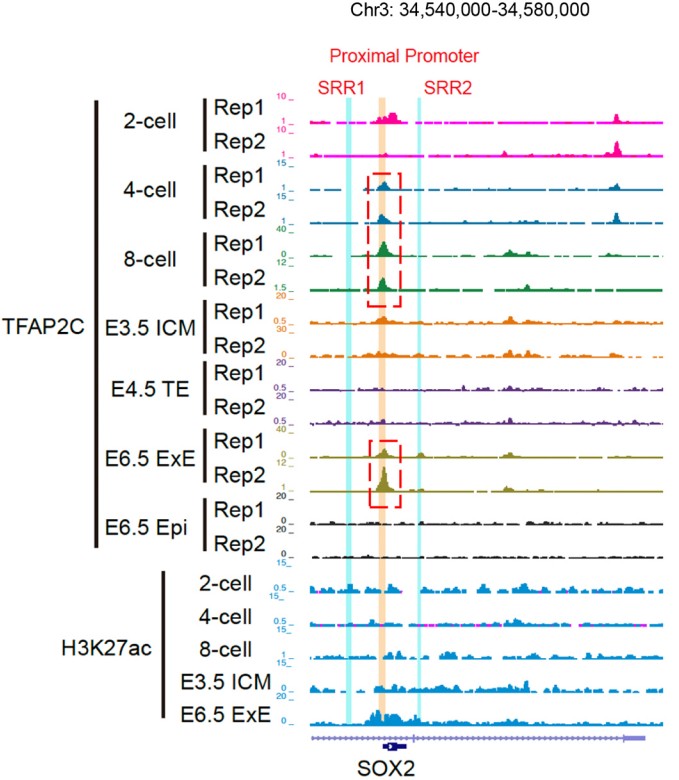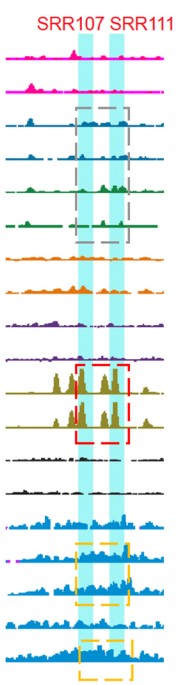

**Fig. 2. TFAP2C preferentially binds to the *Sox2* proximal promoter during preimplantation development.** CUT&RUN analysis of TFAP2C binding to the *Sox2* cis-regulatory elements in preimplantation embryos at the 2-cell, 4-cell, 8-cell and blastocyst stages, and in postimplantation embryos on day 6.5, extra-embryonic ectoderm (ExE) versus epiblast (Epi). TFAP2C binding to the proximal promoter and SRR107/111 is highlighted in the dashed boxes. The red boxes indicate enriched TFAP2C binding. The gray box refers to minimal or no TFAP2C binding. Day 6.5 epiblast is provided as a negative control. H3K27ac enrichment was evaluated in preimplantation embryos and day 6.5 ExE. The orange dashed boxes indicate H3K27ac enrichment.

To examine whether TFAP2C regulates *Sox2* transcription during early embryogenesis, we employed two complementary loss-of-function approaches to disrupt *Tfap2c* expression at the transcriptional and post-transcriptional levels. We first used RNA inference to deplete both maternally and zygotically derived *Tfap2c*

transcripts (Fig. 3A,B). *Tfap2c* or control non-targeting small interfering RNAs (siRNAs) were injected at the 1-cell stage and the embryos were cultured to the 8-cell and morula stages for real-time qPCR analysis and confocal IF analysis. Previously, we demonstrated the specificity and efficacy of the targeting *Tfap2c*

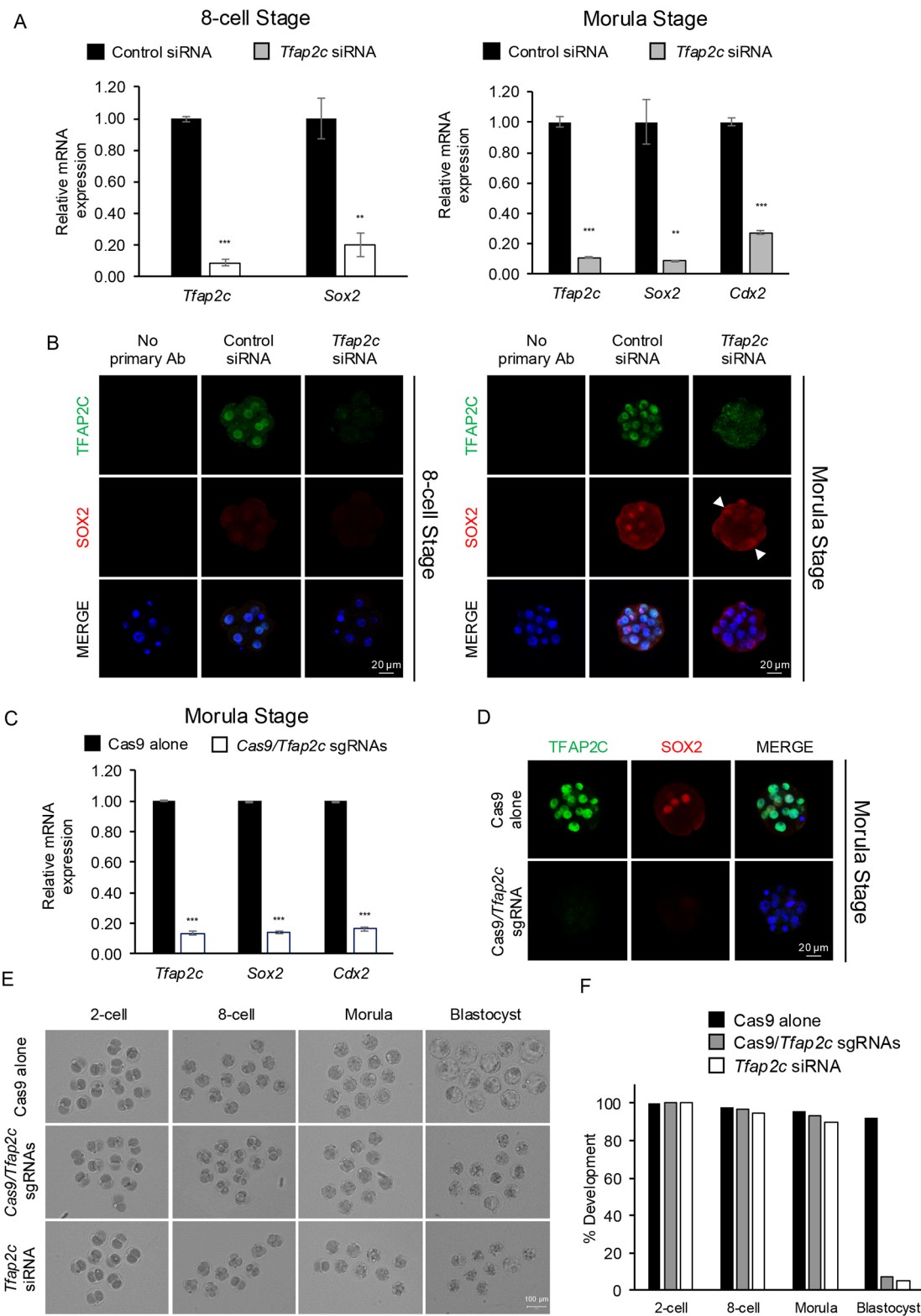

**Fig. 3.** See next page for legend.

**Fig. 3. TFAP2C regulates the onset of zygotic *Sox2* expression in preimplantation embryos.** (A) Real-time PCR analysis of *Tfap2c* and *Sox2* transcripts in embryos injected with control siRNA or *Tfap2c* siRNA. Three biological replicates were used, with 20 pooled 8-cell embryos (E2.5) and morulae (E3.25) per replicate. *Ubtf* was used as a normalization control. Values are presented as mean±s.d. **$P<0.01$, ***$P<0.001$ (paired Student's *t*-test). (B) Confocal IF analysis of TFAP2C and SOX2 in control siRNA- and *Tfap2c* siRNA-injected embryos at the 8-cell (E2.5) and morula stages (E3.25). Nuclei were counterstained with DAPI (blue). Two biological replicates were used for each developmental stage with *n*=5-7 embryos for control siRNA- and *Tfap2c* siRNA-injected embryos. (C) Real-time qPCR analysis of *Tfap2c*, *Sox2* and *Cdx2* transcripts in embryos injected with Cas9 alone or Cas9/*Tfap2c* sgRNA. A total of three biological replicates were used, with 20 pooled morulae (E3.25) per replicate. *Ubtf* was used as a normalization control. Values are presented as mean±s.d. ***$P<0.001$ (paired Student's *t*-test). (D) Confocal IF analysis of TFAP2C and SOX2 in Cas9 alone- and Cas9/*Tfap2c* sgRNA-injected embryos at the morula stage (E3.25). Two biological replicates were used, with *n*=8 control embryos and *n*=23 Cas9/*Tfap2c* sgRNA-injected embryos. *Ubtf* was used as a normalization control. Nuclei were counterstained with DAPI (blue). An equivalent confocal optical section near the equator of each embryo was used for comparisons. (E) Brightfield images of embryos injected with Cas9 alone, Cas9/*Tfap2c* sgRNA or *Tfap2c* siRNA. Embryos were examined at the 2-cell (E1.5), 8-cell (E2.5), morula (E3.25) and blastocyst (E4.5) stages. (F) Percentage of control and treated embryos that developed to the 2-cell, 4-cell, morula and blastocyst stages. Two or three biological replicates were used with a total of 19-57 embryos per group. Scale bars: 20 µm (B,D); 100 µm (E).

siRNA sequences using an elaborate series of control and rescue experiments (Choi et al., 2012). At the 8-cell and morula stages, *Tfap2c* transcripts were reduced by 91 and 89% in *Tfap2c* siRNA-injected embryos, respectively ($P<0.001$; Fig. 3A). Confocal IF analysis revealed undetectable levels of TFAP2C at both stages (Fig. 3B). As a positive control, we evaluated *Cdx2* expression at the morula stage. *Cdx2* is a transcriptional target of TFAP2C in preimplantation embryos (Cao et al., 2015). Consistent with our previous findings, *Cdx2* transcripts were reduced by 73% in *Tfap2c* knockdown (KD) embryos ($P<0.001$; Fig. 3A). Interestingly, evaluation of *Sox2* expression in *Tfap2c* KD embryos revealed that *Sox2* transcripts were reduced by 80 and 91% at the 8-cell and morula stages, respectively ($P<0.01$; Fig. 3A). Most remarkably, SOX2 protein was reduced or not expressed in the inner cells of morula. In many embryos, SOX2 was ectopically expressed in the outer cells, albeit at lower levels (Fig. 3B). We analyzed SOX2 protein localization in control and *Tfap2c* KD morulae by calculating the ratio of SOX2-positive nuclei to the total number of DAPI-positive cells. This analysis revealed a greater ratio of SOX2-positive nuclei to total cells in *Tfap2c* KD morulae compared to control morulae (0.59 versus 0.27, respectively; $P<0.05$; Fig. S5). Furthermore, we used RNA *in situ* hybridization to examine the expression and localization of *Sox2* transcripts in control versus *Tfap2c* KD morulae (Fig. S5). *Sox2* transcripts were enriched in the inner region of control embryos and decreased in *Tfap2c* KD embryos. This finding is concordant with an earlier single-cell gene expression study that revealed that *Sox2* transcripts were enriched within the inner cells of mouse morulae (Guo et al., 2010). These results indicate that TFAP2C may play an important role in the activation and localization of *Sox2* expression in preimplantation embryos.

Next, we employed CRISPR/Cas9 to validate the *Tfap2c* KD phenotype. One-cell-stage embryos were co-injected with Cas9 and two small guide RNAs (sgRNAs) targeting exon 2 to exon 7 of the *Tfap2c* coding region (Fig. S6). The CRISPR-deletion efficiency was confirmed by single embryo genotyping using PCR with outside and inside primers (Fig. S6). This analysis revealed high rates of homozygous targeting in embryos (5/5; 100%). Using this

approach, two subsets of experiments were performed. In the first set of experiments, we injected embryos with either Cas9 alone or Cas9/sgRNAs. Embryos were cultured to the morula stage and analyzed using real-time qPCR analysis and confocal IF. In the Cas9/sgRNA-injected embryos, there was an 86% decrease in *Tfap2c* transcripts and an 86 and 84% reduction in *Sox2* and *Cdx2* transcripts, respectively ($P<0.001$; Fig. 3C). Importantly, TFAP2C protein was completely absent in 23 out of 25 (92%) embryos and SOX2 was not expressed in the inner cells of most embryos (Fig. 3D). Next, we injected embryos with Cas9 alone, Cas9/ sgRNA or *Tfap2c* siRNA and cultured all three groups of embryos to the blastocyst stage to assess development. Consistent with multiple published reports (Choi et al., 2012; Cao et al., 2015; Zhu et al., 2020; Gao et al., 2024), depletion of TFAP2C by either CRISPR or siRNA resulted in high rates of developmental arrest around the morula stage (86% and 84%, respectively, versus 4% of control embryos) (Fig. 3E,F). Only 7 and 5.3% of Cas9/sgRNA- and *Tfap2c*-siRNA-injected embryos, respectively, developed to the blastocyst stage versus 92% of control embryos. Collectively, these results demonstrate that both siRNA and CRISPR/Cas9 are effective approaches for investigating the role of TFAP2C in *Sox2* expression in mouse preimplantation embryos.

## Forced TFAP2C expression induces activation of *Sox2* expression at the 2-cell stage

We next sought to test the hypothesis that TFAP2C is both necessary and sufficient for *Sox2* transcription. To accomplish this, we microinjected either 10 or 25 ng/µl of *Tfap2c* cRNA into 1-cell embryos and cultured them to the 2-cell stage for *Sox2* analysis. Our rationale for examining *Sox2* expression at the 2-cell stage was that zygotic *Sox2* is normally not expressed until the 8-cell stage, and if TFAP2C is a bona fide activator of *Sox2* transcription it will induce the early expression of *Sox2*. The expression of *Cdx2* was evaluated as a positive control (Cao et al., 2015). Accordingly, we observed that *Tfap2c* cRNA was efficiently translated into protein in these experiments (Fig. 4B). Forced expression of TFAP2C significantly induced early *Sox2* expression in a dose-dependent manner. Relative to control embryos, *Sox2* transcripts were induced by ~2- and 6-fold in embryos injected with 10 and 25 ng/µl of *Tfap2c* cRNA, respectively ($P<0.01$; Fig. 4A). Consistent with these results, SOX2 protein was significantly induced in embryos injected with 10 and 25 ng/µl of *Tfap2c* cRNA and was absent in control embryos (Fig. 4B). *Cdx2* expression was induced by ~6- and 13-fold after injection of 10 and 25 ng/µl of *Tfap2c* cRNA, respectively (Fig. S7). These results provide further evidence that TFAP2C functions as an activator of zygotic *Sox2* expression in preimplantation embryos.

## CRISPR/Cas9-mediated disruption of a subset of TFAP2C binding motifs within the *Sox2* proximal promoter prevents the complete activation of *Sox2* expression

To investigate whether TFAP2C regulates *Sox2* expression via the proximal promoter we re-examined the TFAP2C CUT&RUN profile in preimplantation embryos (Li et al., 2024). Because our initial motif analysis identified nine putative TFAPC2-binding sites spanning the promoter and downstream coding region, we wanted to use a more stringent methodology to identify candidate motifs. To accomplish this, we employed the HOMER (Hypergeometric Optimization of Motif EnRichment) tool (Heinz et al., 2010). This analysis revealed the presence of single TFAP2C motif adjacent to a second motif previously identified by manual analysis. Both of these motifs are located within the TFAP2C enrichment peak. Based on these findings, we designed two sgRNAs targeting

A

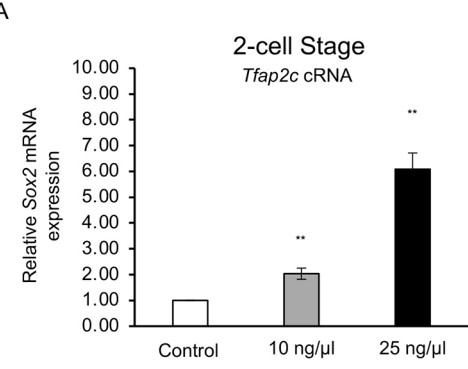

B

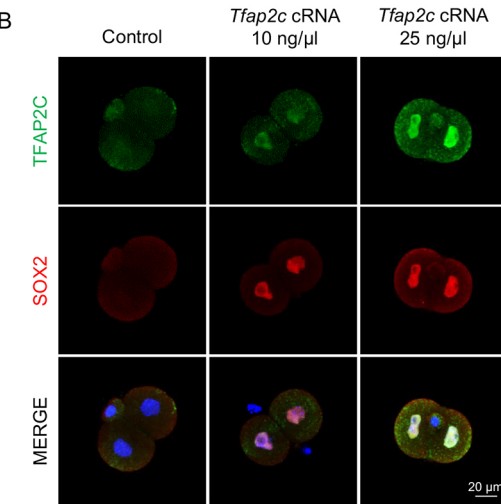

**Fig. 4. Forced TFAP2C expression induces premature *Sox2* expression at the 2-cell stage.** (A) Real-time qPCR analysis of *Sox2* transcripts in control and *Tfap2c* cRNA injected embryos at the 2-cell stage (E1.5). Three biological replicates were used, with 30 pooled embryos per replicate. *Ubtf* was used as a normalization control. Values are mean±s.d. \*\*P<0.01 (paired Student's *t*-test). (B) Confocal IF analysis of TFAP2C and SOX2 in control and *Tfap2c* cRNA-injected embryos at the 2-cell stage (E1.5). Nuclei were counterstained with DAPI (blue). An equivalent confocal optical section near the equator of each embryo was used for comparisons. Three biological replicates were used and a total of 15 embryos from each group were imaged. Scale bar: 20 μm.

the outside sequences of these motifs (Fig. 5A). One-cell embryos were injected with Cas9 alone or Cas9/sgRNAs and then cultured to the morula stage for real-time qPCR and confocal IF analysis. Embryos injected with Cas9/sgRNAs exhibited an ~50% decrease in *Sox2* transcripts (*P*<0.05) and a reduced number of SOX2-positive cells (*P*<0.001; Fig. 5B-D). The ratio of SOX2-positive cells to the total number of cells was reduced in the Cas9/sgRNA-injected embryos compared to embryos that were injected with Cas9 alone (0.06 versus 0.3, respectively; *P*<0.001; Fig. 5E). The CRISPR/Cas9 cutting efficiency in single embryos was confirmed by PCR and gel electrophoresis followed by Sanger sequencing (Fig. S8). In conclusion, these results support a direct role for TFAP2C in the activation of *Sox2* expression during preimplantation embryo development.

### TFAP2C cooperates with the HIPPO signaling pathway to regulate *Sox2* expression

It is well established that *Sox2* expression and localization are controlled by the HIPPO signaling pathway (Wicklow et al., 2014;

Frum et al., 2018, 2019). For example, during the 8-cell to morula transition, HIPPO signaling becomes position dependent and its activity becomes restricted to the inner cells of the embryo by the activity of large tumor suppressor kinase (LATS kinase) (Nishioka et al., 2008, 2009; Karasek et al., 2020). In the inner cells, LATS kinase promotes *Sox2* expression by phosphorylating YAP1 and preventing it from translocating to the nucleus (Wicklow et al., 2014). Conversely, in the outer cells LATS kinase activity is suppressed by the combined actions of apical polarity domain and Rho and Rho-associated coiled-coil kinases 1 and 2 (ROCK1/2) proteins (Karasek et al., 2020). Consequently, YAP1 translocates to the nucleus and forms a complex with TEAD4 to repress *Sox2* expression (Frum et al., 2019). Because both TFAP2C and LATS kinase promote *Sox2* expression during preimplantation development, we sought to examine whether TFAP2C and LATS kinase cooperatively regulate *Sox2* expression.

To address this, we overexpressed wild-type LATS2 in early embryos to downregulate nuclear YAP1 (nYAP1) and augment *Sox2* expression (Nishioka et al., 2009; Frum et al., 2019). One-cell embryos were microinjected with either control siRNA alone, *Tfap2c* siRNA, *Lats2* cRNA, or both *Tfap2c* siRNA and *Lats2* cRNA. Embryos from each group were then cultured to the morula stage for *Sox2* analysis using confocal IF and real-time qPCR. Embryos injected with *Lats2* cRNA displayed a strong reduction in nYAP1 (Fig. 6A) and increased SOX2 expression in both the inner and outer cells (Fig. 6A). qPCR analysis confirmed that *Sox2* transcripts were significantly induced by ~2-fold (*P*<0.05; Fig. 6B). LATS2 overexpression also caused a decrease in *Tfap2c* transcripts, indicating that YAP1 may contribute to *Tfap2c* expression (Fig. 6B). Interestingly, embryos injected with both *Tfap2c* siRNA and *Lats2* cRNA did not exhibit an increase in *Sox2* expression. Instead, these embryos displayed a 74% reduction in *Sox2* transcripts (Fig. 6B). SOX2 protein was reduced and these embryos resembled *Tfap2c* KD embryos (Fig. 6A). Embryos injected with both *Tfap2c* siRNA and *Lats2* cRNA exhibited a strong reduction in nYAP1 (Fig. 6A). Altogether, these results indicate that, at the morula stage, *Sox2* expression requires the presence of the TFAP2C activator and the absence of the YAP1 repressor.

### DISCUSSION

We previously showed that TFAP2C functions as a master regulator of the TE lineage in mouse preimplantation embryos by regulating the expression of genes involved in tight junction biogenesis, cell polarity and TE specification (Choi et al., 2012; Cao et al., 2015). In the present study, we extended these findings and demonstrated that TFAP2C additionally functions as an activator of pluripotency genes, such as *Sox2*. Our findings in preimplantation embryos show that (1) zygotic *Sox2* expression is controlled by the combined actions of a proximal promoter and a distal super enhancer consisting of SRR107 and SRR111; (2) TFAP2C binds to the proximal promoter between the 4-cell and morula stages and is obligatory for proper *Sox2* transcriptional activation; and (3) TFAP2C cooperates with the HIPPO regulator LATS kinase to promote S*ox2* expression.

Our current knowledge on the mechanisms of *Sox2* transcriptional activation comes from genetic and epigenetic studies in ESCs and other cell types (Sikorska et al., 2008; Zhou et al., 2014; Li et al., 2014). As a starting point, we reviewed published literature on mouse ESCs and neuronal cells. We then employed a combination of classic and modern genetic approaches to investigate which ESC cis-regulatory elements are involved in *Sox2* transcriptional activation in mouse preimplantation embryos. Through this analysis, we found that

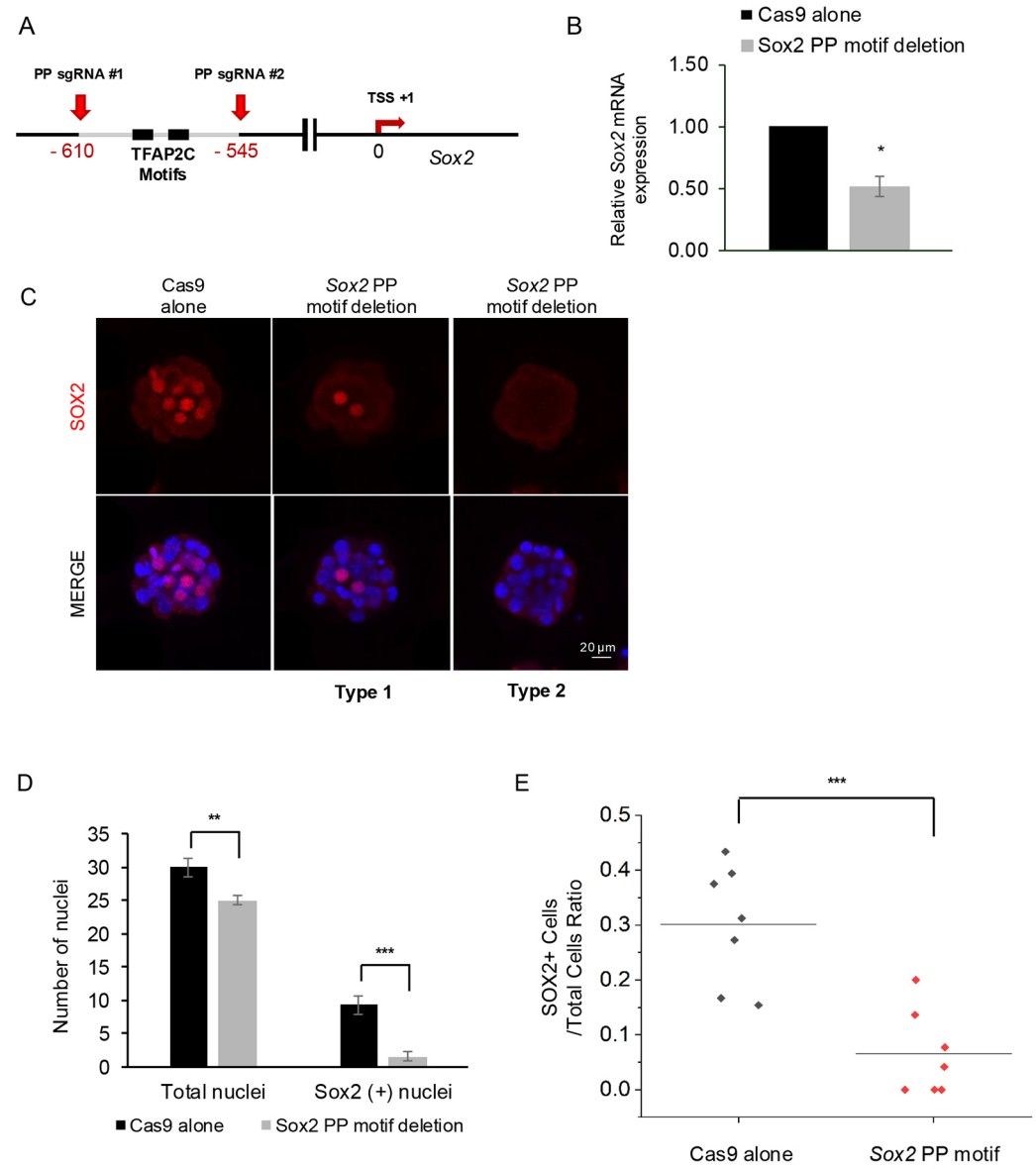

**Fig. 5. Disruption of TFAP2C motifs within the proximal promoter impairs *Sox2* expression.** (A) Schematic illustrating the location of two TFAP2C binding motifs in the *Sox2* proximal promoter (PP). The red arrows indicate the location of the sgRNA binding sites. The two sgRNAs are predicted to delete a ~65 bp fragment containing the TFAP2C motifs. (B) Real-time qPCR analysis of *Sox2* transcripts in control (Cas9 alone) and *Sox2* PP motif-edited embryos at the morula stage (E3.25). A total of three biological replicates were used, with 20 pooled morulae per replicate. Values are presented as mean±s.d. *$P<0.05$ (paired Student's *t*-test). (C) Confocal IF analysis of SOX2 expression in control and *Sox2* PP motif-edited embryos at the late morula stage (E3.75). Two representative embryos are shown. Type 1 embryos exhibited partial SOX2 expression, whereas type 2 embryos exhibited a complete loss of SOX2. A total of three biological replicates were used, with seven embryos per control and treatment groups. Nuclei were counterstained with DAPI (blue). An equivalent confocal optical section near the equator of each embryo was used for comparisons. Scale bar: 20 μm. (D) Quantitation of the total cell numbers and SOX2-positive cell numbers in control versus *Sox2* PP motif-edited embryos at the late morula stage. Values are presented as mean±s.d. **$P<0.01$, ***$P<0.001$ (paired Student's *t*-test). (E) Analysis of the SOX2-positive to total cell number ratios in control versus *Sox2* PP motif-edited embryos. Values are presented as mean±s.d. ***$P<0.001$ (paired Student's *t*-test).

both the *Sox2* promoter and the distal super enhancer (SRR107/SRR111) were required for proper expression. Evaluation of genome-wide H3K27ac data confirmed that SRR107 and SRR111 were active in preimplantation embryos. These findings highlight the conserved nature of these cis-regulatory regions in the regulation of *Sox2* expression in both early embryos and ESCs. Further research is necessary to determine whether there are other cis-regulatory elements specifically involved in the regulation of *Sox2* expression in preimplantation embryos.

Recent work from our laboratory and others revealed that TFAP2C likely functions as an activator of early ICM genes such as *Pou5f1*, *Nanog* and *Nr5a2* (Driscoll et al., 2024a; Li et al., 2024; Zhu et al., 2024). This notion is supported by CUT&RUN, various loss-of-function approaches, and gene expression analysis. In the present study, we re-examined two published TFAP2C binding datasets from preimplantation embryos (Li et al., 2024; Gao et al., 2024). Through this analysis, we found that TFAP2C binding was enriched at the *Sox2* proximal promoter between the 4-cell and

A

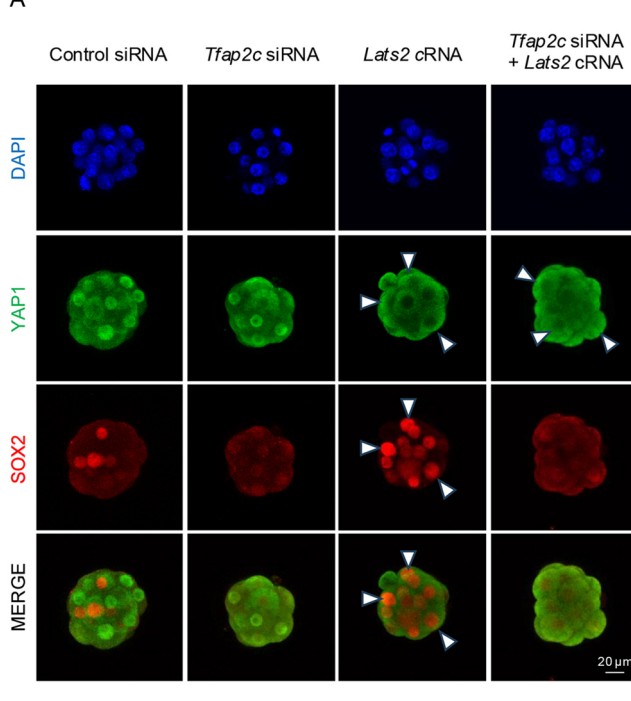

B

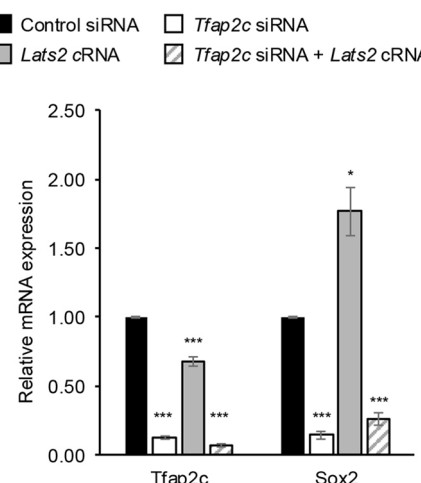

**Fig. 6. TFAP2C cooperates with the HIPPO signaling pathway to regulate *Sox2* expression.** (A) Confocal IF analysis of YAP1 and SOX2 in control siRNA-, *Tfap2c* siRNA- or *Lats2* cRNA-injected, and *Lats2* cRNA and *Tfap2c* siRNA double-injected embryos at the morula stage (E3.25). Two biological replicates were used, with 10-12 embryos per group. An equivalent confocal optical section near the equator of each embryo was used for comparisons. Scale bar: 20 μm. The white arrowheads indicate the nuclear localization of YAP1 or SOX2. (B) Real-time qPCR analysis of *Tfap2c* and *Sox2* transcripts in control siRNA-, *Tfap2c* siRNA- or *Lats2* cRNA-injected, and *Lats2* cRNA and *Tfap2c* siRNA double-injected embryos. Three biological replicates were used, with 20 pooled morula (E3.25) per replicate. *Ubtf* was used as a normalization control. Values are mean±s.d. *$P<0.05$, ***$P<0.001$ (paired Student's *t*-test).

morula stages. These findings are in agreement with a recent study that showed that TFAP2C binding was generally enriched at the promoters of early ICM- and TE-specific genes at the 8-cell stage (Li et al., 2024). Curiously, TFAP2C binding at the super enhancer was minimal during the preimplantation stages of development.

Significant enrichment was not observed until day 6.5 of development in the trophoblast lineage (Li et al., 2024). These observations suggest that TFAP2C-dependent regulation of *Sox2* expression in preimplantation embryos may not involve this enhancer.

Our functional studies in preimplantation embryos support a direct role for TFAP2C in *Sox2* expression. Depletion of TFAP2C impaired the activation of *Sox2* expression between the 8-cell and morula stages, whereas overexpression of TFAP2C induced the premature expression of *Sox2* transcripts and protein at the 2-cell stage. This latter finding is consistent with the role of TFAP2C in cellular reprogramming, where it can induce the expression of repressed genes (Wang et al., 2020; Benchetrit et al., 2015). By using CRISPR/Cas9 to edit the *Sox2* proximal promoter, we determined that there are at least two TFAP2C binding motifs that contribute to the proper activation of *Sox2* expression. Notably, Cas9/sgRNA-injected embryos only exhibited a ~50% reduction in *Sox2* transcripts. Several explanations may account for the incomplete reduction in *Sox2* expression. First, through the manual motif analysis we identified additional binding sites located within the *Sox2* proximal promoter. Because we used CRISPR to edit only two of these motifs, we cannot rule out the importance of the other motifs in *Sox2* expression. Second, our genotyping analysis revealed that the CRISPR/Cas9 cutting efficiency at this genomic region was variable and there was some mosaicism. Some embryonic cells exhibited a complete deletion of the targeted motifs while other cells only had a partial disruption of the binding motifs. The use of suboptimal sgRNAs or the close proximity of the targeting Cas9/sgRNA complexes may have hindered the efficiency of the cutting. Third, it is possible that TFAP2C binding at other known or unknown cis-regulatory elements contributes to *Sox2* expression. Since the CUT&RUN analysis indicated that TFAP2C was weakly enriched at the super enhancer region, we cannot rule out the possibility that this binding is biologically relevant and contributes to *Sox2* regulation in preimplantation embryos. Further experimentation is necessary to address each of these points.

The HIPPO signaling pathway plays a fundamental role in early lineage formation in mammalian preimplantation embryos (Karasek et al., 2020). In mouse embryos, the HIPPO regulator LATS kinase is required for proper ICM lineage specification (Lorthongpanich et al., 2013). For instance, overexpression of LATS kinase promotes an ICM cell fate (Frum et al., 2018), whereas LATS kinase knockdown favors a TE cell fate (Lorthongpanich et al., 2013). In the present study, we overexpressed LATS kinase in early embryos to augment *Sox2* expression. Ablation of TFAP2C in these embryos prevented the LATS-induced increase in *Sox2* expression. This suggests that TFAP2C likely cooperates with LATS kinase to activate *Sox2* expression. This is in alignment with our previous work that showed TFAP2C functions together with the HIPPO signaling pathway to activate the TE regulator *Cdx2* (Cao et al., 2015). In future studies, it will be interesting to manipulate the levels of TFAP2C and LATS kinase in early embryos and somatic cells to examine their potential roles in lineage formation and cellular reprogramming.

SOX2 is a pivotal regulator of lineage formation and pluripotency in mouse embryos (Wicklow et al., 2014; Mistri et al., 2018; Keramari et al., 2010). Our findings offer new clues into how *Sox2* expression may be temporally and spatially regulated during preimplantation embryo development. The early expression pattern of SOX2 is unique compared to other core pluripotency TFs, such as POU5F1 and NANOG, which are ubiquitously expressed up to the mid blastocyst stage (Dietrich and Hiiragi, 2007) before becoming downregulated in the TE cells via different mechanisms (Niwa et al.,

2005; Wang et al., 2010; Paul and Knott, 2014). In the current study, we found that SOX2 protein was downregulated yet ectopically expressed in the outer cells of TFAP2C-deficient morulae. This observation indicates that TFAP2C may play a broader role in *Sox2* regulation by contributing to its proper localization in the inner cells. Future studies are warranted to investigate the specific role of TFAP2C in SOX2 patterning.

TFAP2C represents one of three transcriptional regulators that were recently shown to regulate the transition from totipotency to pluripotent and multipotent cellular states (Driscoll et al., 2024b). TFAP2C is postulated to work in concert with NR5A2 and TEAD4 (e.g. termed TNT) to activate key pluripotency and TE genes, and HIPPO pathway regulators to create a transitory bipotent state at the 8-cell stage (Li et al., 2024; Zhu et al., 2024; Driscoll et al., 2024b). Moreover, TFAP2C and NR5A2 act as pioneer TFs to open chromatin at gene enhancers so SOX2 can bind and regulate gene expression in the ICM (Li et al., 2023). In future studies, it will be important to determine whether NR5A2 and/or TEAD4 are also involved in the activation of *Sox2* expression and its localization. The *Sox2* super enhancer contains numerous TF binding motifs (Zhou et al., 2014), and multiple TFs are likely contributing to its regulation in preimplantation embryos (Zhou et al., 2014). Moreover, it will be exciting to test whether the manipulation of TNT levels in early embryos and totipotent 2C cells accelerates the exit from totipotency and/or promotes a bipotent transitory cellular state that gives rise to both pluripotent and multipotent extra-embryonic stem cells.

In conclusion, the findings reported here provide new insights into the underlying cis-regulatory elements that control *Sox2* expression during mouse preimplantation development. Moreover, they reveal an important role for TFAP2C in the activation of zygotic *Sox2* expression during early lineage formation. These discoveries have important implications in understanding the molecular basis of pluripotency and how conventional trophoblast TFs, such as TFAP2C, contribute to the activation of early pluripotency genes to facilitate a bipotent cellular state that supports lineage formation.

## Limitations

While our study provides new insights into how *Sox2* expression is activated during mouse preimplantation embryo development, there is one weakness that needs to be addressed in future work. The individual cis-regulatory elements that we tested using reporter constructs did not recapitulate the endogenous pattern of SOX2 protein expression. In mice, SOX2 protein is first expressed at the morula stage and restricted to the inside cells by the HIPPO effector YAP1 and TEAD4 (Wicklow et al., 2014; Frum et al., 2019). Injection of our *Sox2* promoter/enhancer H2B-RFP reporter constructs into early embryos induced ubiquitous expression of H2B-RFP at the morula stage. Several explanations may account for this discrepancy. First, our reporter constructs containing only individual cis-regulatory elements may have not been sufficient to restrict H2B-RFP expression to the inside cells. A modified reporter construct containing two or more *Sox2* cis-regulatory elements, such as the proximal promoter and distal enhancers SRR107 and SRR111, may be necessary to fully recapitulate the endogenous expression pattern. Second, it is possible that the cis-regulatory elements we tested in this study are not involved in SOX2 patterning. Other cis-regulatory elements may be required for SOX2 patterning in preimplantation embryos. This opens the door for exciting future research investigations in mouse preimplantation embryos.

## MATERIALS AND METHODS

### Ethics statement

All experiments involving animals were conducted in accordance with the Institutional Animal Care and Use Committee (IACUC) guidelines under current approved protocols at Michigan State University.

### Mouse superovulation, embryo recovery, and *in vitro* culture

Mouse embryos were collected using standard methods. B6D2F1 or CF-1 female mice (6-8 weeks of age) were super-ovulated with 7.5 IU pregnant mare serum gonadotropin (ProSpec Bio), followed by 7.5 IU human chorionic gonadotropin (hCG) 48 h later (Millipore-Sigma). Super-ovulated females were mated with B6D2F1 males and 20 h after hCG injection zygotes were collected from oviducts. Zygotes were released from dissected oviductal ampullae and cumulus cells were removed by pipetting in M2 media supplemented with hyaluronidase. Zygotes were washed three times with M2 media and then cultured in EmbryoMax® KSOM media (Millipore-Sigma) at 37°C in a humidified atmosphere of 5% $CO_2$/5% $O_2$ balanced with $N_2$. Depending on the experiment, embryos were cultured to the following embryonic stages: 2-cell [embryonic day (E) 1.5], 8-cell (E2.5), morula (E3.25), late morula (E3.75), early blastocyst (E4.0) and blastocyst (E4.5).

### Preparation of the *Sox2* reporter assay

The SSRs SRR1, SRR2, SRR107 and SRR111, and a 1.6 kb promoter/partial-CDS were isolated by PCR and cloned into a modified pCl-H2b-RFP vector containing the minimal β-actin promoter (Addgene plasmid #92398). PCR was used to produce linear SRR/PP H2B-RFP constructs for microinjection (Fig. S1). PCR products were purified using a PCR purification kit (QIAGEN).

### Synthesis of *Tfap2c* and *Lats2* cRNA

The coding sequences of *Tfap2c* and *Lats2* were cloned into an *in vitro* transcription plasmid (pIVT) containing 5′ and 3′ *Xenopus* B-globin untranslated regions (UTR) and a poly(A) coding sequence for enhanced translation (Igarashi et al., 2007). Recombinant plasmids were linearized using NdeI and subjected to IVT using mMESSAGE mMACHINE® T7 Ultra Kit (Ambion). After IVT, mRNA was treated with DNase I to remove the DNA template. Purified mRNA was dissolved in RNase-free water and the concentration determined by A260. Synthesis and quality of mRNA were confirmed using gel electrophoresis and a NanoDrop spectrophotometer (Thermo Fisher Scientific).

### siRNA and CRISPR/CAS9-mediated gene targeting

SMART pool siRNA targeting *Tfap2c* mRNA was purchased from Dharmacon (Thermo Fisher Scientific). The *Tfap2c* siRNA was described previously (Choi et al., 2012). CRISPR custom-designed sgRNAs and SpCas9 2NLS nuclease protein were purchased from Synthego. The siRNA and sgRNA sequences are provided in Tables S2 and S3.

### Embryo microinjection

Microinjection was conducted as described previously by our group (Choi et al., 2012). Briefly, injection micropipettes were pulled from micropipette tubes (Drummond) using a Narishige PC-10 puller. The pulled injection needle was cut to generate a 1-2 μm diameter opening. Microinjection was performed using a PL100 picoinjector (Harvard Apparatus) mounted on an Eclipse Ti-U inverted microscope (Nikon). Micromanipulation of embryos was performed in a drop of M2 media covered by mineral oil. For cytoplasmic injections, each embryo was injected with ~5 pl of siRNA, cRNA, or CRISPR reagents at 19-21 h post-hCG injection. For pronuclear injection, ~1-2 pl of linear constructs was injected into the pronucleus at 19-21 h post-hCG. The concentration of the reagents injected into zygotes was 7 ng/μl reporter construct, 50-100 μM siRNAs, 10-25 ng/μl *Tfap2c* cRNA, 200 ng/μl *Lats2* cRNA, 2 pmol sgRNAs and 0.5 pmol CAS9 protein.

### Embryo staging and real-time qPCR

Depending on the experiment, control and manipulated embryos were cultured to the 2-cell (E1.5), 8-cell (E2.5), morula (3.25) or early blastocyst (E4.0) stages for qPCR analysis. Pools of control or manipulated embryos

were selected based on cell number and/or morphology. For each pool of embryos, total RNA was isolated using the PicoPure RNA Isolation Kit (Arcturus). cDNA synthesis was carried out using SuperScript II reverse transcriptase (Invitrogen). Real-time qPCR analysis was conducted utilizing TaqMan probes (Applied Biosystems) or gene-specific designed primers (SYBR Green detection) and a StepOnePlus real-time PCR system (Applied Biosystems). *Ubtf* was used as an endogenous control for embryos as described previously (Wang et al., 2010). The gene-specific Taqman primers used for qPCR are listed in Table S4. SYBR Green PCR master mix (Thermo Fisher Scientific) was used to determine the developmental expression of *Sox2* transcripts. To normalize the expression across different stages of development, *hmGFP* mRNA was added before RNA isolation. The SYBR Green primers are listed in Table S5.

### Embryo staging and confocal IF analysis
Depending on the experiment, control and manipulated embryos were cultured to the 2-cell (E1.5), 8-cell (E2.5), morula (E3.25), late morula (E3.75), early blastocyst (E4.0) and blastocyst (E4.5) stages for IF analysis. At each stage, control and manipulated embryos were matched based on cell number and/or morphology to control for differences. Embryos were fixed using 3.7% formaldehyde in PBS containing 0.1% bovine serum albumin (BSA) for 20 min at room temperature (RT). Fixed embryos were permeabilized with 0.25% Triton X-100 in PBS for 30 min at RT and blocked in 3% BSA overnight at 4°C. Embryos were incubated with the appropriate primary antibodies in blocking solution overnight at 4°C. Embryos treated with the primary antibodies were washed three times in PBS with 0.1% BSA and incubated with secondary antibodies in blocking solution for 30 min at RT. Finally, embryos were transferred to a washing solution containing DAPI (Sigma-Aldrich) for 10 min at RT. Antibodies used for immunostaining were described as follows: rabbit-anti-TFAP2C (PA5-84242, Invitrogen), mouse-anti-TFAP2C (sc-12762, Santa Cruz Biotechnology), goat-anti-SOX2 (GT15098, Neuromics), mouse anti-CDX2 (MU392A-UC, BioGenex), mouse anti-YAP1 (sc-101199, Santa Cruz Biotechnology), rabbit-anti-DsRed (632496, Takara Bio). Images were acquired under an Olympus FluoView 1000 confocal microscope (CARV) with FluoView Viewer 3.0 software (Molecular Devices).

Quantification of SRR/Promoter H2b-RFP reporter constructs was accomplished using ImageJ software (NIH). *z*-section images representing the middle region were used to visualize the entire middle region of each embryo. Images of RFP and DAPI were split into individual channels according to RGB color using the ImageJ program. The staining intensity was measured on both RFP and DAPI images, and the RFP intensity was divided by the DAPI intensity to normalize for nucleus number.

### RNA-fluorescence *in situ* hybridization assay
The RNA-fluorescence *in situ* hybridization (FISH) assay was performed using ViewRNA™ ISH Cell Assay Kit (Invitrogen) following the manufacturer's instructions with some minor modifications for preimplantation embryos. Briefly, embryos were fixed at the morula stage with 3.7% formaldehyde (Avantor) and 0.1% polyvinylpyrrolidone for 20 min. The embryos were washed in PBS containing 0.1% Triton X-100 and 0.1% polyvinylpyrrolidone, and then permeabilized using 1% Triton X-100 in PBS for 30 min. The embryos were washed for 10 min in a detergent solution (from the kit) followed by incubation for 5 min at RT with protease QS (Invitrogen). Lastly, the embryos were washed with wash buffer and incubated at 40°C for 6 h in the probe sets solution containing each target mRNA probe (*Gapdh*, *Sox2*). The embryos were counterstained with DAPI, and images acquired under an Olympus FluoView 1000 confocal microscope (CARV) with FluoView Viewer 3.0 software (Molecular Devices).

### Single embryo genotyping assay
Individual morula or blastocysts were washed with PBS and transferred directly into a PCR tube containing 2 µl of genotyping lysis buffer (0.1% Tween 20, 0.1% Triton X-100 and 4 µg/ml proteinase K). The samples were incubated for 20 min at 56°C, and proteinase K was inactivated by heating at 95°C for 10 min. Lastly, 4 µl of distilled water was added to each sample and samples were stored at −20°C until use. PCR was conducted using 2 µl of gDNA sample with One-Taq DNA polymerase (NEB). In a subset of experiments, the purified PCR products were analyzed by Sanger sequencing (Agenta) and a web-based program Inference of CRISPR Edits (ICE; Synthego) was used to evaluate the modified sequences. All primers used for genotyping are listed in Tables S2, S6 and S8.

### TFAP2C CUT&RUN data analysis
TFAP2C binding at the *Sox2* proximal promoter and SRRs was determined using a TFAP2C CUT&RUN data set (accession number GSE216256; Li et al., 2024). The CUT&RUN procedure was based on a previously published protocol (Skene et al., 2018) and is described in detail by Li et al. The TFAP2C stage-specific binding peaks were identified in 2-cell, 4-cell, 8-cell, E3.5 ICM, E4.5 TE, E6.5 epiblast, and E6.5 (ExE). E6.5 epiblasts were used as a negative control because TFAP2C expression is downregulated at this stage. The UCSC genome browser (Perez et al., 2025) was used to visualize the TFAP2C CUT&RUN signals by generating RPKM values on a 100-bp-window basis.

### Statistical analysis
All of the experiments were conducted at least three times for statistical analysis. Data were either analyzed using paired Student's *t*-test or one-way ANOVA followed by Dunnett's post-hoc test. Data are presented as mean±s.d. $P<0.05$ was considered significant.

### Acknowledgements
We thank Dr Amy Ralston (MSU) for providing us with a protocol for single embryo genotyping. We also thank Dr Bin Gu (MSU) for advice on the CRISPR experiments. We thank the MSU Center for Advanced Microscopy for their support and expertise with the confocal imaging analysis.

### Competing interests
The authors declare no competing or financial interests.

### Author contributions
Conceptualization: J.G.K., J.K.; Data curation: J.G.K., J.K.; Formal analysis: J.G.K., J.K., C.S.D., L.L., W. X.; Funding acquisition: J.G.K.; Investigation: J.G.K., J.K., C.S.D., L.L., C.A.W., W.X.; Methodology: J.G.K., J.K., L.L., C.A.W., W.X.; Project administration: J.G.K.; Supervision: J.G.K., W.X.; Validation: J.G.K., J.K., C.S.D., L.L., W.X.; Visualization: J.G.K., J.K., C.S.D., L.L., W.X.; Writing – original draft: J.G.K., J.K.; Writing – review & editing: J.G.K., J.K., C.S.D., C.A.W., W.X.

### Funding
This work was supported by a grant from the National Institute of Child Health and Human Development (HD095371 to J.G.K.) and Michigan State University AgBioResearch. C.S.D. was supported by a T32 doctoral fellowship from the National Institute of Child Health and Human Development (HD087166). Open Access funding provided by Michigan State University. Deposited in PMC for immediate release.

### Data and resource availability
All relevant data and details of resources can be found within the article and its supplemental information.

### Peer review history
The peer review history is available online at https://journals.biologists.com/dev/lookup/doi/10.1242/dev.204626.reviewer-comments.pdf

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
