## [Peer Review File · Development (Cambridge, England)]

Deciphering the role of cis-regulatory elements and TFAP2C in the activation of zygotic Sox2 expression in mouse preimplantation embryos

Jaehwan Kim, Chad S. Driscoll, Lijia Li, Catherine A. Wilson, Wei Xie and Jason G. Knott
DOI: 10.1242/dev.204626

Editor: Peter Rugg-Gunn

Review timeline

Original submission:	30 December 2024
Editorial decision:	27 January 2025
First revision received:	21 May 2025
Editorial decision:	16 June 2025
Second revision received:	23 June 2025
Accepted:	25 June 2025

Original submission

First decision letter

MS ID#: dev.204626

MS TITLE: Role of genomic regulatory regions and TFAP2C in the initiation and cell-specific expression of zygotic Sox2 in early mouse embryos

AUTHORS: Jason G. Knott; Jaehwan Kim; Chad Schoen Driscoll; Lijia Li; Catherine Wilson; Wei Xie

Dear Dr Knott,

I have now received all the referees' reports on the above manuscript, and have reached a decision. The referees' comments are appended below, or you can access them online: please go to:

As you will see, the referees express interest in your work, but have some significant criticisms and recommend a substantial revision of your manuscript before we can consider publication. If you are able to revise the manuscript along the lines suggested, I will be happy to receive a revised version of the manuscript. Your revised paper will be re-reviewed by one or more of the original referees, and acceptance of your manuscript will depend on your addressing satisfactorily the reviewers' major concerns. Please also note that Development will normally permit only one round of major revision. If it would be helpful, you are welcome to contact us to discuss your revision in greater detail.

Please attend to all of the reviewers' comments and ensure that you clearly highlight all changes made in the revised manuscript. Please avoid using 'Tracked changes' in Word files as these are lost in PDF conversion. I should be grateful if you would also provide a point-by-point response detailing how you have dealt with the points raised by the reviewers in the 'Response to Reviewers' box. If you do not agree with any of their criticisms or suggestions please explain clearly why this is so.

Reviewer 1

SUMMARY OF THE ADVANCE MADE IN THIS PAPER AND ITS POTENTIAL SIGNIFICANCE TO THE FIELD

This study investigated the mechanisms of Sox2 expression in preimplantation mouse embryos. The authors first identified genomic regulatory elements required for expression at the morula stage. They then focused on TFAP2C, a regulator of trophoblast and bipotency, and demonstrated that TFAP2C binds to the promoter region at the bipotent 8-cell stage. TFAP2C is essential for Sox2 expression at the 8-cell and morula stages and induces Sox2 expression at the 2-cell stage. At the morula stage, TFAP2C is also required for the inside-restricted expression of SOX2. Although the identification of Sox2 regulatory sequences and the discovery of their regulation by TFAP2C are novel, the analyses lack sufficient depth, and the relationships among Sox2 cis-regulatory elements, TFAP2C, and Sox2 expression remain unclear. Furthermore, the involvement of the trophoblast transcription factor TFAP2C in the activation of early pluripotency genes, Nanog and Pou5f1, has been recently reported, and is therefore not conceptually novel.

SUGGESTIONS TO AUTHORS

The relationships among Sox2 cis-regulatory elements, TFAP2C, and Sox2 expression are not clear. The following points need improvement:

- (1) The identified regulatory elements, proximal promoter (PP), SRR107, and SRR111, induce ubiquitous reporter expression at the morula stage, which does not reproduce the inside-restricted expression of Sox2. While SRR107 and SRR111 are required for strong expression, their specific roles in Sox2 regulation remain unclear.
- (2) The timing of analyses varies among experiments. Enhancer activity and endogenous gene expression were primarily analyzed at the morula stage, while TFAP2C CUT & RUN data demonstrated TFAP2C binding to PP at the 8-cell stage. The 8-cell stage represents the bipotency state, while the morula stage corresponds to ICM-fate specification regulated by Hippo signaling. To elucidate regulatory mechanisms in the bipotency state, enhancer/expression studies should be conducted at the 8-cell stage. Conversely, to reveal regulatory mechanisms at the morula stage, TFAP2C CUT & RUN data from this stage are necessary.
- (3) Although the authors discussed the importance of TFAP2C binding to PP, the functional significance of this binding was not experimentally validated. The model presented in Figure 6 states, "The first mechanism involves direct binding of TFAP2C to the PP and SE, which induces Sox2 expression between the 2-cell and 8-cell stage." However, the activities of PP and SE at the 8-cell stage have not been examined.

Figure S1. The details of the reporter constructs are unclear. Does IRS2 refer to IRES2? For the PP construct, was the PP fragment inserted upstream of the β -actin promoter as described in the figure legend? Similarly, line 132 states, "H2b-RFP alone did not exhibit RFP fluorescence," but it is unclear whether this construct lacks a promoter or includes the β -actin promoter but lacks the enhancer fragment. Adding a schematic diagram summarizing the reporter constructs would help the readers' understanding.

Line 145 and Figure S2. "The CRISPR-deletion efficiency was confirmed by single embryo genotyping using PCR at the blastocyst stage." The deletion efficiency should be described numerically for each deletion experiment. Figure S2B,C shows amplification of the wildtype band in some lanes. The presence of genomic DNA in each sample should be confirmed with positive-control experiments, such as PCR amplification of an unrelated genomic region.

Figure S3. It is unclear whether the expressions of TFAP2C and SOX2 overlap at the morula and blastocyst stages. This should be addressed by showing a single confocal section of immunofluorescence-stained embryos. It is impossible to determine cell positions within the embryo (inside or outside) from Z-stacked images. The differentiation status of ICM cells differs significantly between E3.5 and E4.5 blastocysts. Which stage of embryos was used as blastocysts in this study?

Line 254 and Figure S5. "The CRISPR-deletion efficiency was confirmed by single embryo genotyping at the morula stage." The deletion efficiency should be described numerically in the text.

Figure 4. Why was SOX2 expression analyzed at the 2-cell stage? All analyses should be performed at the same developmental stage; otherwise, interpretation of the results becomes difficult.

Injecting RNA into one blastomere of the two-cell-stage embryos and analyzing at the 8-cell stage, as performed in Zhu et al. (2024), would be a more suitable experimental approach.

Figure 5C, D. The data show an interaction between TFAP2C and YAP1. However, the role of this interaction is not clarified in this paper. Although the authors proposed a model in Figure 6, stating, "TFAP2C may form a complex with nYAP1 in the outer cells to directly repress Sox2 expression," this is purely speculative. I suggest the following experiment to clarify whether TFAP2C is involved in Sox2 repression by TEAD4-nYAP1: Compare SOX2 expression between Lats1/2 knockdown (via siRNA injection) embryos and Lats1/2; Tfap2c double-knockdown embryos. The former embryos should exhibit strong suppression of SOX2 throughout the embryo via TEAD4-nYAP1. If TFAP2C is required for this suppression, then the latter embryos should express SOX2.

All statistical analyses were performed with Student's t-test. However, for comparisons involving multiple groups, one-way ANOVA with appropriate post-hoc tests should be used.

Reviewer 2

In this manuscript, Kim and colleagues explore the transcriptional regulation of Sox2 in the mouse preimplantation embryo. This is an interesting issue, and the results presented here contribute to the still limited description of the *in vivo* activity of cis-regulatory elements in the early mouse embryo.

Certainly, the cis-regulation of Sox2 in pluripotent ESCs has already been described (Zhou 2014, Li 2014), and the results presented here regarding the Sox2 SE elements SRR107 and SRR111 mainly corroborate these observations. The authors then suggest that TFAP2C regulates Sox2 through these enhancer elements. TFAP2C has already been shown to regulate both TE and ICM genes, but the evidence that in this case it does so directly through the Sox2 SE is rather weak (see below). Finally, the authors build a model of mutual interaction between Hippo and Sox2 (what equally has already been shown in the embryo) together with Tfap2c, that is rather speculative.

The following issues should be addressed:

1. The demonstration of the sufficiency of SE elements SRR107 and SRR111 to drive expression in the embryo, and their necessity for Sox2 expression, is convincingly shown in Fig. 1B, C, and a strong point of the manuscript. However, the statement that these elements are necessary for lineage determination is not so well grounded. This is based solely on the co-staining of SOX2 and CDX2 in deleted embryos. However, using SOX2 as a marker of the ICM lineage when its own expression is being affected by the deletion of some of its enhancers, is not a strong argument. Furthermore, the changes in ratios of SOX2+ and CDX2+ populations shown in Fig. 1F, although statistically significant, are not very pronounced. Is the total cell number maintained in deleted embryos? This should be easy to see. More evidence would be needed to support the necessity of the enhancers in lineage specification. An option would be to look at other markers (OCT4, NANOG, GATA3?), or examine more advanced embryos.
2. Regarding the deleted embryos, the genotyping strategy that has been used does not allow distinguishing heterozygotes from wild types. Also, is lack of a PCR product enough strong evidence for identifying a homozygous deletion? Couldn't primers external to the deletion be used as positive proof? Also, it would be good to include a positive control for DNA integrity in each genotype embryo. The SRR107 F and R primers could be used in the SRR111 deleted embryos and vice versa. It is true that the amount of genomic DNA that can be obtained from a single preimplantation embryo is very limited, but it might be worth a try.
3. Next, the authors explore the input of TFAP2c on the described Sox2 enhancers. TFAP2C binding is assessed using previously published CUT&RUN data. Contrary to what is stated in the text (line 201), there is no binding to SRR107 and very low binding to SRR111, compared to that in E6.5 ExE (by the way, are these enhancer elements active in the ExE at those stages? It would be nice to see). This data is not sufficient to state that TFAP2C bind to these elements and through them regulate Sox2 expression.

4. Tfap2c siRNAs lead to strong reduction of Sox2 as measured by qPCR at both 8-cell and morula stages (Fig. 3A). However, the immunos shown in Fig. 3B show that in morulas, SOX2 levels are similar although with a redistribution between inner and outer cells, as mentioned in this section and also further along when describing Fig. 5. This contradiction should be addressed and better explained.
5. Here, a minor point. The image shown for a control siRNA injection in Figs 3B and Fig. 5A is the same morula. It is not very important, but the authors should mention it or change the image in one of the panels. Another option is to group the data from Figs 3 and 5 in a single figure, as it is the same observation (although taking into account the contradiction mentioned above).
6. The authors show in Fig. 3E, F that knockdown or knockout of Tfap2c leads to lethality between morula and blastocyst stages. This is in contradiction with the work of Li et al. (2024), as these authors find that maternal-zygotic Tfpa2c KO embryos make normal blastocysts and survive up to E6.5. Therefore, morulas injected with reagents to deplete TFPA2C must be compromised and not very healthy. Could this explain why the authors find lack of expression of Sox2 and Cdx2? Is any other lineage marker (e.g., OCT4, GATA3, NANOG) still expressed correctly at these stages? It might be worth to check some of them, and also other ways to assess if these embryos are still alive or already dead (for example, proliferation or measuring active transcription).
7. The PLA assay shown in Fig. 5D is interesting, but adds little more than would a double immuno showing co-localization. In any case, a quantification of this experiment should be provided.
8. The data regarding the interactions between TFAP2C, Hippo and Sox2 shown in Fig. 5E is, to say the least, confusing and in some way redundant. The effect of Tfap2c knockdown on SOX2 is shown here for the third time in the manuscript, together with a not very clear change to nuclear YAP1. Lats2 overexpression leads to reduced nuclear YAP1, as expected, and upregulation of SOX2, as is already known. And finally, the combination of both gives the same result as Tfap2c knockdown alone. The simplest explanation for this data is that Tfap2c regulates Sox2 independently of Hippo.
9. Finally, the model in Fig. 6 is quite speculative and not based on the data shown here. In first place, the diagrams of embryos in the top row are difficult to understand. As for the model of transcriptional regulation shown below (left panel), the authors have not convincingly demonstrated that binding of TFAP2C to the SE regulates Sox2 expression. The CUT&RUN data shown in Fig. 2 indicates rather little direct binding (see above), and only experiments where gain and loss of function of Tfap2c injected together with the RFP enhancer constructs (wt and versions with putative TFPA2C binding sites mutated) would allow to say so. On the other hand, the right panel apparently suggests that the putative interaction between TFAP2C, YAP1 and TEAD4 only occurs on the Sox2 PP, and not the SE. Is this what the authors want to say? Overall, this model and the discussion should at least be clear on the lack of strong evidence to support it.

Reviewer 3

SUMMARY OF THE ADVANCE MADE IN THIS PAPER AND ITS POTENTIAL SIGNIFICANCE TO THE FIELD

In this manuscript by Kim et al, the authors provide a good analysis of the regulation of Sox2 during early embryogenesis. They identify important regulatory elements surrounding Sox2 for its expression and *in vivo* and convincingly show that Tfpa2c is a major regulator, directly but also through the Hippo pathway. This is a good study shedding mechanistic light on an important period of embryogenesis. I have some comments that the authors should consider to strengthen some of the manuscript's conclusions, as detailed below.

SUGGESTIONS TO AUTHORS

1/ I am confused by the reporter assays described: as I understand them, they test for transcriptional activity using enhancer elements in the absence of the endogenous Sox2 proximal promoter (PP), which is independently tested. If true, then it may be important to assess the activity of the PP when linked to the different enhancers under study (especially because SRR2 is not that bad in inducing expression).

2/ The analysis of Sox2 and Cdx2 immunostaining upon Crispr-mediated deletions could be more convincing if the authors could show an analysis independent of qualifying the cells as positive for one or the other marker, that is, present a scatter plot of Sox2 vs Cdx2 expression for each analyzed cell.

3/ Binding of TFAP2C at the PP is very robust and very well documented in Fig2. However, this is much less the case for the super-enhancer, even at 8c stage where the binding profiles are quite inconsistent between replicates. Asking for additional Cut&Run would be the ideal option, but if the authors can comment about this issue or clarify in the text that the main strong target of TFAP2C during pre-implantation development is the PP perhaps it would be enough.

4/ The results showing morula arrest upon si/sgRNA-mediated depletion of Tfp2c are important. It would however be better if the authors could show that the overexpression they also implement can rescue these deficits, such that it is fully established that it is the loss of Tfp2c the underlying cause.

5/ There is a semantic/conceptual issue in the beginning of the introduction, in the link between the first and second sentence: if the zygote is already totipotent, then the capacity to differentiate in embryonic and extraembryonic cannot be acquired during the first cell fate decisions. Either the zygote is not totipotent, or, if it is, the differentiation potential cannot be acquired later. It is possible to argue to both options, but the others should select one and stick to it to ensure full logical consistence in their views.

6/ It is unclear whether Sox2 can be considered the first pluripotency factor expressed in inner cells of the morula (line 82) - other pluripotency TFs are already expressed. Do they authors refer to first pluripotency TF expressed in inner cells specifically, and not in outer cells? If so, please clarify; if not, please correct. This is also important given the more nuanced view given later, lines 285/286

First revision

Author response to reviewers' comments

Dear Reviewers:

We are grateful for your constructive criticisms that helped us improve the overall quality of the manuscript. The manuscript is now more focused and provides new data that strengthen the central conclusions. We improved the Discussion and included a Limitations section discussing the shortcomings of our reporter construct studies. We have also slightly revised the manuscript title and used the proper terminology to describe genomic regulatory regions (*i.e.*, cis-regulatory elements). Below we address each of your comments point-by-point. Our responses are shown in **blue font**. We hope that our responses and the revised manuscript are now satisfactory for publication in *Development*. Thank you for your time and efforts in reviewing our manuscript.
Professor Knott

REVIEWER 1

1) The identified regulatory elements, proximal promoter (PP), SRR107, and SRR111, induce ubiquitous reporter expression at the morula stage, which does not reproduce the inside-restricted expression of Sox2. While SRR107 and SRR111 are required for strong expression, their specific roles in Sox2 regulation remain unclear.

Author response:

- Thank-you for this comment. We now thoroughly address this point in the Discussion under the Limitations section.

Limitations

While our study provides new insights into how *Sox2* expression is activated during mouse preimplantation embryo development, there is one weakness that needs to be addressed in future work. The individual cis-regulatory elements that we tested using reporter constructs did not recapitulate the endogenous pattern of SOX2 protein expression. In mice, SOX2 protein is first expressed at the morula stage and restricted to the inside cells by the HIPPO effector YAP1 and TEAD4 (Wicklów et al., 2014, Frum et al., 2019). Injection of our *Sox2* promoter/enhancer H2B-RFP reporter constructs into early embryos induced ubiquitous expression of H2B-RFP at the morula stage. Several explanations may account for this discrepancy. Firstly, our reporter constructs containing only individual cis-regulatory elements may have not been sufficient to restrict H2B-RFP expression to the inside cells. A modified reporter construct containing two or more *Sox2* cis-regulatory elements, such as the proximal promoter and distal enhancers, SRR107 and SRR111, may be necessary to fully recapitulate the endogenous expression pattern. Secondly, it is possible that the cis-regulatory elements we tested in this study are not involved in SOX2 patterning. Other novel cis-regulatory elements may be required for SOX2 patterning in preimplantation embryos. This opens the door for exciting future research investigations in mouse preimplantation embryos.

2) The timing of analyses varies among experiments. Enhancer activity and endogenous gene expression were primarily analyzed at the morula stage, while TFAP2C CUT & RUN data demonstrated TFAP2C binding to PP at the 8-cell stage. The 8-cell stage represents the bipotency state, while the morula stage corresponds to ICM-fate specification regulated by Hippo signaling. To elucidate regulatory mechanisms in the bipotency state, enhancer/expression studies should be conducted at the 8-cell stage. Conversely, to reveal regulatory mechanisms at the morula stage, TFAP2C CUT & RUN data from this stage are necessary.

Author response:

- Thank-you for the constructive comment. The majority of our experiments focused on studying *Sox2* regulation at the morula stage because *Sox2* expression is significantly greater at this timepoint. In preliminary experiments we used real-time PCR analysis to determine the timing of *Sox2* mRNA expression. This analysis revealed that *Sox2* mRNA expression initiates at the middle 8-cell stage, but its expression is significantly lower than the morula stage. This finding is supported by publicly available transcriptome data at NCBI. Thus, for the majority of our studies we focused on the morula stage when *Sox2* mRNA expression is higher and SOX2 protein is first detected. Based on these findings, we analyzed the expression of the reporter constructs at the morula stage. To check if RFP expression was induced at earlier stages of development we went back to our embryo data files. We found some slower growing embryos at the early morula stages (8-16 cells). RFP expression was visible in these embryos, indicating that reporter constructs were activated at earlier stages. Below are examples of embryos at the 8-16 cells stages.

- The CUT&RUN data from Li et al revealed that TFAP2C binds to the *Sox2* proximal promoter at the 8-cell stage when transcription begins. Because this dataset does not include morula, we examined a second dataset from Gao et al using the UCSC genome browser. This analysis indicated that TFAP2C is enriched at the *Sox2* promoter at the morula stage before it is lost at the blastocyst stage (Li et al., 2024). These data support a role for TFAP2C in regulating *Sox2* expression at the 8-cell and morula stages. In the revised manuscript we now provide additional text in the results and discussion sections.

3) Although the authors discussed the importance of TFAP2C binding to PP, the functional significance of this binding was not experimentally validated. The model presented in Figure 6 states, "The first mechanism involves direct binding of TFAP2C to the PP and SE, which induces *Sox2* expression between the 2-cell and 8-cell stage." However, the activities of PP and SE at the 8-cell stage have not been examined.

Author response:

- Thank-you for this very important point and encouraging us to perform additional experiments. We now provide new data supporting the functional significance of TFAP2C binding to the *Sox2* proximal promoter. In the original manuscript we performed a manual motif analysis of the *Sox2* promoter and partial coding region. This analysis identified 9 potential TFAP2C motifs. Because of the large number of potential motifs, we were originally discouraged from using CRISPR to edit these candidates. However, we reanalyzed this promoter/partial coding region using a more stringent method (HOMER) and the UCSC genome browser. Accordingly, we identified two TFAP2C motifs located ~500 bp upstream of the transcriptional start site (TSS). Importantly, these motifs are located within the TFAP2C enrichment peak. Using CRISPR/Cas9 and a pair of sgRNAs flanking the 5- and 3' sequences of the motifs, we discovered that these two motifs are important for the proper activation of *Sox2* expression at the early morula stage. These new data are presented in Figure 5 and Fig. S8. The technical and biological limitations of the experiment are discussed in the Discussion section.

4) Figure S1. The details of the reporter constructs are unclear. Does IRS2 refer to IRES2? For the PP construct, was the PP fragment inserted upstream of the β -actin promoter as described in the figure legend? Similarly, line 132 states, "H2b-RFP alone did not exhibit RFP fluorescence," but it is unclear whether this construct lacks a promoter or includes the β -actin promoter but lacks the enhancer fragment. Adding a schematic diagram summarizing the reporter constructs would help the readers' understanding.

Author response:

- We apologize for the confusion. We now include a detailed schematic diagram summarizing each reporter construct that was tested in preimplantation embryos. The 1.6kb promoter/partial-CDS fragment was tested using a H2b-RFP construct lacking the minimal β -actin promoter. We accidentally referred to this construct as the proximal

promoter in the original manuscript. We better describe this construct now. Moreover, the H2b-RFP construct alone (negative control) only contained the β -actin promoter without any additional enhancers. IRS2 refers to IRES2.

5) Line 145 and Figure S2. "The CRISPR-deletion efficiency was confirmed by single embryo genotyping using PCR at the blastocyst stage." The deletion efficiency should be described numerically for each deletion experiment. Figure S2B,C shows amplification of the wildtype band in some lanes. The presence of genomic DNA in each sample should be confirmed with positive-control experiments, such as PCR amplification of an unrelated genomic region.

Author response:

- For each CRISPR experiment we now describe the deletion efficiency numerically in the results section. Unfortunately, we did not have leftover genomic DNA from the original embryo injection experiments to rerun a positive control region. Our remaining NIH funding for this project is very limited and we did not want to exhaust funds on repeating the many CRISPR embryo microinjection experiments to verify the PCR reactions. We are very confident in our original PCR genotyping results using outside and inside primers at each targeted region. We were able to get consistent amplification of the wild-type samples when ran in parallel with the CRISPR/Cas9 targeted embryos. Importantly, we obtained very strong protein phenotypes for each of the targeted genes and enhancer regions. For example, when we used CRISPR and two sgRNAs to target exons 2-7 of *Tfap2c*, we observed a complete loss of TFAP2C protein in 23/25 (92%) of the embryos. Because *Tfap2c* is biallelically expressed, heterozygous targeting would not eliminate the protein expression. Thus, we are confident that in our CRISPR experiments in Figures 1 and 3 we obtained high rates of homozygous targeting. Below we provide examples of the two types of TFAP2C protein phenotypes we observed in CRISPR/Cas9 injected embryos.

6) Figure S3. It is unclear whether the expressions of TFAP2C and SOX2 overlap at the morula and blastocyst stages. This should be addressed by showing a single confocal section of immunofluorescence-stained embryos. It is impossible to determine cell positions within the embryo (inside or outside) from Z-stacked images. The differentiation status of ICM cells differs significantly between E3.5 and E4.5 blastocysts. Which stage of embryos was used as blastocysts in this study?

Author response:

- Thank-you for this important critique. In the supplemental results (now Fig.S4 A,B) we provide two sets of confocal immunofluorescent (IF) images for TFAP2C and SOX2. We include both single confocal sections and Z-stack images of the same embryos at the 2-

cell, 8-cell, morula, and blastocyst stages. The single sections (Fig. S4B) better highlight the inside-specific expression of SOX2 protein at the morula and blastocyst stages. Confocal IF was performed on *in vitro* cultured (IVC) blastocysts on day 4.5 of development. We chose this timepoint, because IVC blastocysts grow slower than *in vivo* derived blastocysts (day 3.5) and undergo blastocyst formation on day 4 to 4.5.

7) Line 254 and Figure S5. "The CRISPR-deletion efficiency was confirmed by single embryo genotyping at the morula stage." The deletion efficiency should be described numerically in the text.

Author response:

- For each CRISPR experiment in Figure 1 and Figure 3 we now describe the deletion efficiency numerically in the results section.

8) Figure 4. Why was SOX2 expression analyzed at the 2-cell stage? All analyses should be performed at the same developmental stage; otherwise, interpretation of the results becomes difficult. Injecting RNA into one blastomere of the two-cell-stage embryos and analyzing at the 8-cell stage, as performed in Zhu et al. (2024), would be a more suitable experimental approach.

Author response:

- We apologize for not clearly explaining the logic behind this experiment. We now provide the rationale for examining *Sox2* expression at the 2-cell stage. We previously attempted a similar approach as Zhu et al. but quickly determined that it was more difficult to interpret the results because of the high levels of endogenous *Tfap2c* expression at the 8-cell and morula stages. In the results section we now include the following rationale for focusing our analysis on the 2-cell stage. "Our rationale for examining *Sox2* expression at the 2-cell stage was that zygotic *Sox2* is normally not expressed until the 8-cell stage, and if TFAP2C is a bona fide activator of *Sox2* transcription it will induce the early expression of *Sox2*." In the Discussion section we now discuss the significance and implications of these findings.

9) Figure 5C, D. The data show an interaction between TFAP2C and YAP1. However, the role of this interaction is not clarified in this paper. Although the authors proposed a model in Figure 6, stating, "TFAP2C may form a complex with nYAP1 in the outer cells to directly repress *Sox2* expression," this is purely speculative. I suggest the following experiment to clarify whether TFAP2C is involved in *Sox2* repression by TEAD4-nYAP1: Compare SOX2 expression between *Lats1/2* knockdown (via siRNA injection) embryos and *Lats1/2*; *Tfap2c* double-knockdown embryos. The former embryos should exhibit strong suppression of SOX2 throughout the embryo via TEAD4-nYAP1. If TFAP2C is required for this suppression, then the latter embryos should express SOX2.

Author response:

- We apologize for the confusion and the over interpretation of the data in the original manuscript. Thank you for the insightful comments and suggestions. We attempted the experiment you suggested and did not observe the predicted phenotype. We performed two biological replicates and analyzed 8 embryos in each group. SOX2 protein expression was reduced in both TFAP2C KD and TFAP2C/LATS Kinase KD embryos (See the confocal images below). Based on your comments and those of reviewer two, we have significantly revised this section of the manuscript. Because our data don't support a role for TFAP2C and YAP1 in repressing *Sox2* expression in the outside cells, we removed the *Lats kinase* siRNA and TFAP2C and YAP1 PLA data. We agree with reviewer two that our data only supports a role for TFAP2C in acting upstream (independently) of the HIPPO signaling to activate *Sox2* expression. We focused on this finding in the revised manuscript. We have also removed the model from the manuscript. The overall key findings are summarized in the first paragraph of the discussion.

10) All statistical analyses were performed with Student's t-test. However, for comparisons involving multiple groups, one-way ANOVA with appropriate post-hoc tests should be used.

Author response:

- We apologize for the oversight. Thank you for this important suggestion. For comparisons involving multiple groups we performed one-way ANOVA followed by the Dunnett's test for corrections. The data in Figure 1 (C, D, E, F) now contain updated statistics. The results section were updated to reflect this analysis.

REVIEWER 2

1) The demonstration of the sufficiency of SE elements SRR107 and SRR111 to drive expression in the embryo, and their necessity for Sox2 expression, is convincingly shown in Fig. 1B, C, and a strong point of the manuscript. However, the statement that these elements are necessary for lineage determination is not so well grounded. This is based solely on the co-staining of SOX2 and CDX2 in deleted embryos. However, using SOX2 as a marker of the ICM lineage when its own expression is being affected by the deletion of some of its enhancers, is not a strong argument. Furthermore, the changes in ratios of SOX2+ and CDX2+ populations shown in Fig. 1F, although statistically significant, are not very pronounced. Is the total cell number maintained in deleted embryos? This should be easy to see. More evidence would be needed to support the necessity of the enhancers in lineage specification. An option would be to look at other markers (OCT4, NANOG, GATA3?), or examine more advanced embryos.

Author response:

- Thank you for the constructive comments. We agree that our current enhancer data does not provide strong evidence for a role in lineage specification. We have revised the manuscript accordingly and removed that conclusion throughout. Based on the current data we can only conclude that disruption of these Sox2 enhancer regions altered lineage allocation in early blastocysts. We used CDX2 as a marker of the TE cells because its expression was not negatively impacted by deletion of the Sox2 enhancers. Total cells numbers, based on DAPI staining, were similar amongst groups. We did not stain for OCT4 and NANOG because these proteins are ubiquitously expressed up to the mid-blastocyst stage before being downregulated in the TE cells. Moreover, OCT4 and NANOG would not serve as a reliable markers of the ICM/epiblast in later stage blastocysts because SOX2 regulates their expression at this stage of development (Wicklow et al., 2014). For example, in SOX2 null blastocysts OCT4 and NANOG expression are undetectable.

2) Regarding the deleted embryos, the genotyping strategy that has been used does not allow distinguishing heterozygotes from wild types. Also, is lack of a PCR product enough strong evidence for identifying a homozygous deletion? Couldn't primers external to the deletion be used as

positive proof? Also, it would be good to include a positive control for DNA integrity in each genotype embryo. The SRR107 F and R primers could be used in the SRR111 deleted embryos and vice versa. It is true that the amount of genomic DNA that can be obtained from a single preimplantation embryo is very limited, but it might be worth a try.

Author response:

- Thank you for the comment and suggestions. Unfortunately, we did not have leftover genomic DNA from the original embryo injection experiments to rerun a positive control region. Our remaining funding for this project is very limited and we did not want to exhaust funds on repeating the many CRISPR embryo microinjection experiments to verify the PCR reactions for SRR107, SRR111, SCR, and *Tfap2c* exons 2-7. We are very confident in our original PCR genotyping results using outside and inside primers at each targeted region. We were able to get consistent amplification of the wild-type samples when ran in parallel with the CRISPR/Cas9 targeted embryos. We agree that a positive control reaction would have been a good control. In the revised manuscript we include a new CRISPR experiment that supports a role for TFAP2C in regulating *Sox2* expression via the proximal promoter. In this experiment we genotyped single embryos using external primers spanning the edited region, as you recommended.
- We think that the lack of a PCR product is strong enough evidence for homozygous targeting. For example, when we used CRISPR and two sgRNAs to target exons 2-7 of *Tfap2c*, we observed a complete loss of TFAP2C protein in 23/25 (92%) of the embryos. Because *Tfap2c* is biallelically expressed, heterozygous targeting would not have eliminated the protein expression. Thus, we are confident that we obtained high rates of homozygous targeting. Please see the additional confocal data provided to reviewer one (comment 5).

3) Next, the authors explore the input of TFAP2c on the described *Sox2* enhancers. TFAP2C binding is assessed using previously published CUT&RUN data. Contrary to what is stated in the text (line 201), there is no binding to SRR107 and very low binding to SRR111, compared to that in E6.5 ExE (by the way, are these enhancer elements active in the ExE at those stages? It would be nice to see). This data is not sufficient to state that TFAP2C bind to these elements and through them regulate *Sox2* expression.

Author response:

- Thank you for the insightful and constructive comments. We have revised the results section accordingly and better describe the TFAP2C binding profile at the *Sox2* distal enhancers (SRR107 and SRR111) in preimplantation embryos versus day 6.5 postimplantation embryos. We agree that there is very little TFAP2C binding at SRR107 and SRR111 in preimplantation embryos. It's unclear whether this minimum binding is biologically relevant. We now discuss this point in the Discussion section. The entire manuscript was revised, and it now focused on the role of TFAP2C binding at the *Sox2* proximal promoter in preimplantation embryos.
- Based on your additional comment above, we evaluated the H3K27ac dataset in Li et al., 2024. We observed H3K27ac enrichment at SRR107 and SRR111 in 4-cell and 8-cell stage embryos and day 6.5 ExE. These data further support that these enhancers are active in preimplantation embryos and early postimplantation stage embryos. The TFAP2C CUT&RUN data provided in Figure 2 are now updated to include the H3K27ac profiles.

4) *Tfap2c* siRNAs lead to strong reduction of *Sox2* as measured by qPCR at both 8-cell and morula stages (Fig. 3A). However, the immunos shown in Fig. 3B show that in morulas, SOX2 levels are similar although with a redistribution between inner and outer cells, as mentioned in this section and also further along when describing Fig. 5. This contradiction should be addressed and better explained.

Author response:

- Thank you for the comment. Depletion of *TFAP2C* by siRNA and CRISPR/Cas9 induces a strong reduction in *Sox2* mRNA. When evaluating SOX2 protein expression by confocal IF in both types of *TFAP2C* deficient embryos, we observed more background in the *Tfap2c* siRNA injected embryos stained with SOX2 antibody. For example, the cytoplasmic signal appeared brighter in these embryos. In our experience sometimes siRNA injected embryos, irrespective of the gene knockdown, exhibit higher levels of background. Oppositely, in CRISPR/Cas9 injected embryos SOX2 protein was greatly reduced and there was less cytoplasmic background.
- Another potential explanation is that siRNA only provides a temporary reduction in *TFAP2C* that influences SOX2 expression. Closer examination of *Tfap2c* siRNA morula indicate that there is some faint staining of *TFAP2C* protein in the nuclei. Because a single injection of siRNA at the 1-cell stage will eventually become diluted with each subsequent cell division, its plausible that some *TFAP2C* protein will be recovered by the morula stage. This is not the case for CRISPR/Cas9 edited embryos because Cas9 and the sgRNAs introduce edits in the 1-cell stage embryos that induce continuous suppression of *Tfap2c*.

5) Here, a minor point. The image shown for a control siRNA injection in Figs 3B and Fig. 5A is the same morula. It is not very important, but the authors should mention it or change the image in one of the panels. Another option is to group the data from Figs 3 and 5 in a single figure, as it is the same observation (although taking into account the contradiction mentioned above).

Author response:

- Thank you for this comment. We mistakenly used the same control embryo as a representative image in both figures. We now use two separate control embryos as representative images in Fig. 3 and Fig. 5. Note: the original data provided in Fig. 5A was moved to the supplemental data files (Fig.S5).

6) The authors show in Fig. 3E, F that knockdown or knockout of *Tfap2c* leads to lethality between morula and blastocyst stages. This is in contradiction with the work of Li et al. (2024), as these authors find that maternal-zygotic *Tfap2c* KO embryos make normal blastocysts and survive up to E6.5. Therefore, morulas injected with reagents to deplete *TFAP2C* must be compromised and not very healthy. Could this explain why the authors find lack of expression of *Sox2* and *Cdx2*? Is any other lineage marker (e.g., OCT4, GATA3, NANOG) still expressed correctly at these stages? It might be worth to check some of them, and also other ways to assess if these embryos are still

alive or already dead (for example, proliferation or measuring active transcription).

Author response:

- Author response redacted for reasons of confidentiality

7) The PLA assay shown in Fig. 5D is interesting but adds little more than would a double immuno showing co-localization. In any case, a quantification of this experiment should be provided.

Author response:

- Thank you for the comment. Based on the reviewer comments we have removed these data from the original figure that is now Figure 6.

8) The data regarding the interactions between TFAP2C, Hippo and Sox2 shown in Fig. 5E is, to say the least, confusing and in some way redundant. The effect of Tfp2c knockdown on SOX2 is shown here for the third time in the manuscript, together with a not very clear change to nuclear YAP1. Lats2 overexpression leads to reduced nuclear YAP1, as expected, and upregulation of SOX2, as is already known. And finally, the combination of both gives the same result as Tfp2c knockdown alone. The simplest explanation for this data is that Tfp2c regulates Sox2 independently of Hippo.

Author response:

- Thank you very much for the constructive comments. We agree that these data were confusing and that we overinterpreted the results. In response to reviewer one we attempted an additional experiment that did not produce meaningful results. Thus, we have significantly revised this section of the Results and modified the conclusions in the Discussion. We now conclude that TFAP2C acts upstream (independently) of LATS kinase to positively regulate Sox2 expression. This finding consistent with our earlier work that showed TFAP2C functions upstream of the HIPPO signaling pathway to activate *Cdx2* expression (Cao et al., 2015).

9) Finally, the model in Fig. 6 is quite speculative and not based on the data shown here. In first place, the diagrams of embryos in the top row are difficult to understand. As for the model of transcriptional regulation shown below (left panel), the authors have not convincingly demonstrated that binding of TFAP2C to the SE regulates Sox2 expression. The CUT&RUN data shown in Fig. 2 indicates rather little direct binding (see above), and only experiments where gain and loss of function of Tfp2c injected together with the RFP enhancer constructs (wt and versions with putative TFPA2C binding sites mutated) would allow to say so. On the other hand, the right panel apparently suggests that the putative interaction between TFAP2C, YAP1 and TEAD4 only occurs on the Sox2 PP, and not the SE. Is this what the authors want to say? Overall, this model and the discussion should at least be clear on the lack of strong evidence to support it.

Author response:

- Thank for your constructive comments and suggestions for improvement. We have significantly revised the manuscript and removed that speculative model. We agree that TFAP2C binding to the super enhancer is minimal during preimplantation embryo development. This observation is consistent with published work showing TFAP2C generally binds to the promoters of ICM and TE genes in the preimplantation embryos. We now provide new data supporting a role for TFAP2C in Sox2 expression through the proximal promoter. Our initial studies in Figure 1 and Figure S1 and S2 tested which known cis-regulatory elements are required for Sox2 expression in mouse preimplantation embryos. Next, we focused on the role of TFAP2C in Sox2 activation and the importance of the Sox2 proximal promoter. Our overall key findings are summarized in the first paragraph of the discussion.

REVIEWER 3:

1) I am confused by the reporter assays described: as I understand them, they test for transcriptional activity using enhancer elements in the absence of the endogenous Sox2 proximal promoter (PP), which is independently tested. It true, then it may be important to assess the activity of the PP when linked to the different enhancers under study (especially because SRR2 is not that bad in inducing expression).

Author response:

- Thank you for the helpful comments. We now provide a schematic diagram in Fig. S1 describing the features of each reporter construct. For all of the enhancers tested, we used the minimum β -actin promoter. We now better explain each construct in the manuscript text. For example, the Sox2 promoter refers to a 1.6kb promoter/partial coding region that spans the transcriptional start site. We clearly describe this promoter reporter construct now. When we tested this fragment, we removed the β -actin promoter from the construct.

2) The analysis of Sox2 and Cdx2 immunostaining upon Crispr-mediated deletions could be more convincing if the authors could show an analysis independent of qualifying the cells as positive for one or the other marker, that is, present a scatter plot of Sox2 vs Cdx2 expression for each analyzed cell.

Author response:

- Thank you, for the suggestion. We reanalyzed these data and presented them as a scatter plot. We now provide these as supplemental data in Fig. S3.

3) Binding of TFAP2C at the PP is very robust and very well documented in Fig2. However, this is much less the case for the super-enhancer, even at 8c stage where the binding profiles are quite inconsistent between replicates. Asking for additional Cut&Run would be the ideal option, but if the authors can comment about this issue or clarify in the text that the main strong target of TFAP2C during pre-implantation development is the PP perhaps it would be enough.

Author response:

- Thank you for your comment. We have revised the results section accordingly and better describe the TFAP2C binding profile at the Sox2 proximal promoter and distal enhancers (SRR107 and SRR111) in preimplantation embryos. We agree that there is very little TFAP2C binding at SRR107 and SRR111 in preimplantation embryos. The entire manuscript was revised, and it now focuses on TFAP2C binding at the proximal promoter in preimplantation embryos.

4) The results showing morula arrest upon si/sgRNA-mediated depletion of Tfp2c are important. It would however be better if the authors could show that the overexpression they also implement can rescue these deficits, such that it is fully established that it is the loss of Tfp2c the underlying cause.

Author response:

- Thank you for this comment. We previously addressed this point in a previous publication (Choi et al., 2012). In brief, we conducted rescue experiments where *Tfp2c* cRNA and *Tfp2c* siRNA were microinjected into early embryos. We were able to significantly restore blastocyst development, and the expression *Tfp2c* target genes identified in that study. Note: In all of our subsequent papers including this manuscript, we have used the same *Tfp2c* targeting siRNA sequences obtained by Dharmacon.

5) There is a semantic/conceptual issue in the beginning of the introduction, in the link between the first and second sentence: if the zygote is already totipotent, then the capacity to differentiate in embryonic and extraembryonic cannot be acquired during the first cell fate decisions. Either the zygote is not totipotent, or, if it is, the differentiation potential cannot be acquired later. It is possible to argue to both options, but the others should select one and stick to

it to ensure full logical consistence in their views.

Author response:

- Thank you for this comment. We have simplified those sentences and removed the term totipotent when describing the early embryo that is undergoing the first cell-fate decisions.

6) It is unclear whether Sox2 can be considered the first pluripotency factor expressed in inner cells of the morula (line 82) - other pluripotency TFs are already expressed. Do they authors refer to first pluripotency TF expressed in inner cells specifically, and not in outer cells? If so, please clarify; if not, please correct. This is also important given the more nuanced view given later, lines 285/286

Author response:

- Thank you for this comment. We agree that several pluripotency TFs are expressed in the early mouse embryo. However, SOX2 is the first pluripotency factor that is restricted/localized to the inner cells of the morula. We now clarify this point in the manuscript text.

Second decision letter

MS ID#: dev.204626R1

MS TITLE: Deciphering the role of cis-regulatory elements and TFAP2C in the activation of zygotic Sox2 expression in mouse preimplantation embryos

AUTHORS: Jason G. Knott; Jaehwan Kim; Chad Schoen Driscoll; Lijia Li; Catherine Wilson; Wei Xie

Dear Dr Knott,

I have now received all the referees reports on the above manuscript, and have reached a decision. The referees' comments are appended below, or you can access them online: please go to .

The overall evaluation is positive and we would like to publish a revised manuscript in Development, provided that the remaining referees' comments can be satisfactorily addressed (Reviewers 1 and 3). Please attend to their comments in your revised manuscript and detail them in your point-by-point response. If you do not agree with any of their criticisms or suggestions explain clearly why this is so.

Reviewer 1

SUMMARY OF THE ADVANCE MADE IN THIS PAPER AND ITS POTENTIAL SIGNIFICANCE TO THE FIELD

This study investigated the mechanisms underlying Sox2 expression in preimplantation mouse embryos through in vivo enhancer analyses. The authors first showed that the key cis-regulatory elements include a proximal promoter and distal super enhancer. They also showed that the binding of TFAP2C, a regulator of trophectoderm and bipotency, to the proximal promoter is a critical mechanism for Sox2 activation. Finally, they also showed that TFAP2C and the Hippo signaling pathway cooperatively regulate Sox2. The identification of Sox2 regulatory sequences and their regulation by TFAP2C is novel. Although these regulatory sequences are not sufficient to reproduce spatial expression pattern of Sox2, their findings provide important clues to understand the molecular mechanisms by which ICM fate and/or pluripotency are established in preimplantation embryos.

SUGGESTIONS TO AUTHORS

In the revised manuscript, the authors have appropriately addressed most of the issues raised by this reviewer. I only have relatively minor comments, which do not require additional experiments. I recommend the authors consider these points before publication in *Development*, particularly the one regarding the interpretation of results.

Comments

Lines 209-211:

In this section, the authors state: "we examined a second CUT&RUN data set which included morula (Gao et al., 2024). This analysis revealed that TFAP2C binding was enriched at the Sox2 proximal promoter in 8-cell and morula stage embryos (data not shown)." Given the importance of this result, the data should be included in Figure 2 rather than omitted.

Page 11:

Based on a set of Tfap2c knockdown and LATS2 overexpression experiments, the authors concluded that TFAP2C functions upstream of LATS kinase. However, this interpretation is misleading. The data does not support a regulatory relationship between TFAP2C and LATS kinase; rather, the two factors appear to act in parallel. The results indicate that Sox2 expression requires two conditions: the presence of the TFAP2C activator and the absence of the TEAD-YAP repressor. Therefore, a more accurate interpretation is that TFAP2C and the Hippo signaling pathway cooperatively regulate Sox2 expression.

Similar misinterpretations appear in the abstract and discussion sections, where the authors suggest a hierarchical relationship between TFAP2C and LATS kinase. These sections should be revised to reflect the parallel and cooperative nature of their roles in regulating Sox2.

Reviewer 2

The authors have properly addressed the concerns raised in the first revision of the manuscript, providing new data or convincingly arguing about the difficulties in addressing some of the issues. The revised version has been thoroughly re-written, and all the cases where there was some over-interpretation of the data has been corrected. Now, the manuscript represents a solid piece of work that deals with the identification of transcriptional responses during mouse preimplantation development, and as such is a worthy addition to the field.

Reviewer 3

The authors have addressed all of my comments and the paper is now almost suitable for publication. I find the replies to other referees also of good quality, overall. I would insist, however, to further modify Fig.S3 to show the proper immunostaining quantifications rather than the fraction of +ive cells.

Second revision

Author response to reviewers' comments

Dear Reviewers:

Thank you for the additional comments to further improve the quality of our manuscript. Below we have addressed each of your comments point-by-point. Our responses are in **blue font**. We hope that our responses and the revised manuscript are now satisfactory for publication in *Development*. Thank you again for your time and efforts in reviewing our manuscript.

Best regards,

Jason Knott

REVIEWER 1

In the revised manuscript, the authors have appropriately addressed most of the issues raised by this reviewer. I only have relatively minor comments, which do not require additional experiments. I recommend the authors consider these points before publication in *Development*, particularly the one regarding the interpretation of results.

Author response: Thank you so much for thoroughly reviewing our revision and providing additional helpful suggestions. Please see our responses below.

Lines 209-211:

In this section, the authors state: "we examined a second CUT&RUN data set which included morula (Gao et al., 2024). This analysis revealed that TFAP2C binding was enriched at the Sox2 proximal promoter in 8-cell and morula stage embryos (data not shown)." Given the importance of this result, the data should be included in Figure 2 rather than omitted.

Author response: Author response partly redacted for reasons of confidentiality. Rather than omit these data, we now include them in Fig. S3.

Page 11:

Based on a set of Tfp2c knockdown and LATS2 overexpression experiments, the authors concluded that TFAP2C functions upstream of LATS kinase. However, this interpretation is misleading. The data does not support a regulatory relationship between TFAP2C and LATS kinase; rather, the two factors appear to act in parallel. The results indicate that Sox2 expression requires two conditions: the presence of the TFAP2C activator and the absence of the TEAD-YAP repressor. Therefore, a more accurate interpretation is that TFAP2C and the Hippo signaling pathway cooperatively regulate Sox2 expression. Similar misinterpretations appear in the abstract and discussion sections, where the authors suggest a hierarchical relationship between TFAP2C and LATS kinase. These sections should be revised to reflect the parallel and cooperative nature of their roles in regulating Sox2.

Author response: Thank you so much for this constructive comment. We agree with you that this is a more accurate interpretation of our results. We have revised the interpretation and/or conclusions throughout the manuscript as recommended. We now state that TFAP2C and the HIPPO signaling pathway act cooperatively to regulate Sox2 expression.

REVIEWER 2

The authors have properly addressed the concerns raised in the first revision of the manuscript, providing new data or convincingly arguing about the difficulties in addressing some of the issues. The revised version has been thoroughly re-written, and all the cases where there was some over-interpretation of the data has been corrected. Now, the manuscript represents a solid piece of work that deals with the identification of transcriptional responses during mouse preimplantation development, and as such is a worthy addition to the field.

Author response: Thank you so much for your positive comments and feedback. We are delighted you find the manuscript now suitable for publication in *Development*.

REVIEWER 3

The authors have addressed all of my comments, and the paper is now almost suitable for publication. I find the replies to other referees also of good quality, overall. I would insist,

however, to further modify Fig.S3 to show the proper immunostaining quantifications rather than the fraction of +ive cells.

Author response: Thank you for carefully reviewing our revised manuscript and providing additional feedback. We apologize for not completely understanding how you wanted us to present the lineage allocation data in Fig. 1F. We were unsure if you were suggesting that we quantitate the relative levels of SOX2 or CDX2 protein in single cells using a method such as Image J. It is not feasible to do this in blastocysts because of the large number of cells. The original way we presented the data in Fig. 1F is consistent with many publications in the field. For example, to determine the impact of a specific gene knockout or gene knockdown on ICM/TE lineage allocation, researchers typically present the data as the number of OCT4, NANOG, or SOX2-positive cells (ICM markers) to the total number of cells in blastocysts (Zhao et al., 2024; Cui et al., 2020; Zhu et al., 2024) . Likewise, the number of CDX2-positive cells (TE marker) to the total number of cells is used for the TE lineage (Simmet et al., 2018; Cui et al., 2020; Zhao et al., 2024). These data are generally presented using a table or either a bar graph or dot plot. We have removed the scatterplot presentation of these data in Fig. S3 and have kept the original bar graph representation in Fig. 1F. We appreciate all of your suggestions throughout the review process.

BLASTOCYST LINEAGE ALLOCATION REFERENCES:

CUI, W., CHEONG, A., WANG, Y., TSUCHIDA, Y., LIU, Y., TREMBLAY, K. D., MAGER, J. 2018 MCRS1 is essential for epiblast development during early mouse embryogenesis. *Reproduction*, 1, 1-13.

SIMMET, K., ZAKHARTCHENKO, V., PHILIPPOU-MASSIER, J. BLUM, H., KLYMIUK, N., WOLF, E. 2018. OCT4/POU5F1 is required for NANOG expression in bovine blastocysts. *PNAS*, 11, 2770-2775.

ZHAO, Y., ZHANG, M., LIU, J., HU, X., SUN, Y., HUANG, X., LI, J., LEI, L. 2024. Nr5a2 ensures inner cell mass formation in mouse blastocyst. *Cell Rep*, 3, E113840.

ZHU, M., MEGLICKI, M., LAMBA, A., WANG, P., ROYER, C., TURNER, K., JAUHAR, M. A., JONES, C., CHILD, T., COWARD, K., NA, J. & ZERNICKA-GOETZ, M. 2024. Tead4 and Tfap2c generate bipotency and a bistable switch in totipotent embryos to promote robust lineage diversification. *Nat Struct Mol Biol*, 31, 964-976.

Third decision letter

MS ID#: dev.204626R2

MS TITLE: Deciphering the role of cis-regulatory elements and TFAP2C in the activation of zygotic Sox2 expression in mouse preimplantation embryos

AUTHORS: Jason G. Knott; Jaehwan Kim; Chad Schoen Driscoll; Lijia Li; Catherine Wilson; Wei Xie
ARTICLE TYPE: Research Article

Dear Dr Knott,

I am happy to tell you that your manuscript has been accepted for publication in *Development*, pending our standard publication integrity checks.